# Gene-informed decomposition model predicts lower soil carbon loss due to persistent microbial adaptation to warming

Xue Guo (iD) et al.#

Soil microbial respiration is an important source of uncertainty in projecting future climate and carbon (C) cycle feedbacks. However, its feedbacks to climate warming and underlying microbial mechanisms are still poorly understood. Here we show that the temperature sensitivity of soil microbial respiration ($Q_{10}$) in a temperate grassland ecosystem persistently decreases by 12.0 ± 3.7% across 7 years of warming. Also, the shifts of microbial communities play critical roles in regulating thermal adaptation of soil respiration. Incorporating microbial functional gene abundance data into a microbially-enabled ecosystem model significantly improves the modeling performance of soil microbial respiration by 5–19%, and reduces model parametric uncertainty by 55–71%. In addition, modeling analyses show that the microbial thermal adaptation can lead to considerably less heterotrophic respiration (11.6 ± 7.5%), and hence less soil C loss. If such microbially mediated dampening effects occur generally across different spatial and temporal scales, the potential positive feedback of soil microbial respiration in response to climate warming may be less than previously predicted.

#A list of authors and their affiliations appears at the end of the paper.

Soil stores large quantities of organic carbon (C), about three times more C than the Earth's atmosphere[1,2]. Soil respiration is the largest single source of carbon dioxide ($CO_2$) from terrestrial ecosystems to the atmosphere, and is about ten times larger than anthropogenic emissions[3]. Soil total respiration ($R_t$) includes both autotrophic respiration ($R_a$) from plant root growth and root biomass maintenance, and heterotrophic respiration ($R_h$) from microbial decomposition of litter and soil organic matter (SOM). Various short-term experiments show that soil respiration increases exponentially with temperature[4], which has been used as a general relationship to parameterize ecosystem and Earth System Models (ESMs)[5]. If the near-exponential short-term relationship of soil respiration and temperature holds for the long-term (years to decades), climate warming will trigger a sharp increase in ecosystem respiration (ER). Such an increase could then result in a strong positive feedback to the global C cycle[6], which is dependent on the responses of $R_h$ and the dynamics of detrital inputs under warming[7]. Therefore, it is particularly important to accurately evaluate soil $R_h$ and its response to climate warming. However, partitioning $R_t$ into $R_a$ and $R_h$ is one of the main challenges in both experiment- and model-based global change research[8]. Consequently, soil respiration is a poorly understood key C flux in the global C cycle and is an important source of the uncertainty in climate projections[9–11].

Microorganisms can dramatically adjust their respiratory responses to temperature over long terms (years) via changing their metabolism and community structure[12]. Several climate change experiments demonstrated that soil respiration was stimulated in the short term, followed by a dampened effect of warming later[13–15]. This phenomenon is referred to as thermal adaptation of soil respiration[16,17]. The existence of thermal adaptation of soil respiration is of critical importance as the greater the global thermal adaptation of soil respiration, the weaker the positive feedback between climate warming and ecosystem $CO_2$ release[18]. However, the existence and the degree of thermal adaptation of soil respiration is extremely uncertain, especially in the field and over a long duration (years to decades)[9,10,19]. Whether thermal adaptation of soil respiration can persist over time is not clear. Moreover, the mechanisms controlling thermal adaptation of soil respiration have been intensively debated[4,14,19–21], and include warming-induced substrate depletion[19,21] or evolutionary adaptation via individual acclimatization and changes in microbial community[13,14]. These two mechanisms may lead to different soil C loss in a warmer world[14,21]. While the former could lead to a depletion of labile C pools, releasing more C into the atmosphere through microbial respiration if more plant-derived C is available under warming, the latter could result in less soil labile C loss due to microbial community adaptation to the rising temperature (warming)[14]. Therefore, knowledge about thermal adaptation of soil respiration and its underlying mechanisms will be central to making better predictions of terrestrial C cycling feedbacks. However, one grand challenge in climate change biology is to integrate microbial community information, particularly omics information, into ecosystem models to improve their predictive ability for projecting future climate and environmental changes[22]. More specifically, parameter values for various microbial processes are poorly constrained by experimental observations, which becomes one of the significant uncertainty sources leading to low confidence in carbon-climate feedback projections[23]. Hence, using omics-enabled experimental observations to improve model parameter estimations could greatly help to refine the projected magnitude of the carbon-climate feedbacks.

Soil microbial communities are very complex in structure and are sensitive to changes in environmental conditions[14], so information obtained from a single time point provides only a snapshot of the microbial community, and is not suitable for ecosystem model simulation. To model microbial respiratory responses to climate warming, long-term experiments under more realistic field settings with time-series microbial data are needed. Otherwise, it will be difficult to determine the direction, magnitude, and duration of biospheric feedbacks to climate change[15,24]. Therefore, a new warming experiment site with sandy soil and dominance of $C_3$ grasses was established in a native, tall-grass prairie ecosystem of the US Great Plains in Central Oklahoma (34° 59′N, 97° 31′W) in July 2009[25]. The warmed plots were subjected to continuous warming by infrared radiators (+3 °C), and annual soil samples were archived over subsequent years and analyzed by integrated metagenomics technologies.

In this study, we examine the temperature responses of soil $R_h$ (7 years) and their underlying mechanisms. Our main objectives are to answer the following questions: (i) How does long-term experimental warming affect the temperature responses of soil microbial respiration over time? (ii) Whether or not thermal adaptation of microbial respiration occurs persistently across years under warming and by what underlying mechanisms? (iii) Can the microbial mechanisms underlying soil respiration be incorporated into ecosystem models to improve model performance and reduce model uncertainty? Our study reveals that thermal adaptation of microbial respiration exists persistently over the long-term and that the shifts of microbial communities play critical roles in regulating such thermal adaptation of microbial respiration. Incorporating metagenomics-based microbial functional genes significantly increases confidence in model simulations, indicating that the microbial thermal adaptation could lead to considerably less heterotrophic respiration and hence less soil C loss.

## Results and discussion

**Overall ecosystem changes under long-term warming.** The plots in the warming experiment site have been subjected to continuous warming for over 7 years[7]. On average, experimental warming significantly ($p < 0.01$) increased daily air temperature by 1.3 °C, and daily mean soil temperature at 7.5 cm by 2.8 °C (Fig. 1a). Experimental warming significantly ($p < 0.01$) decreased soil moisture by 6.4% (Fig. 1b). Consistent with previous reports[14], warming significantly ($p = 0.01$) shifted plant community structure. Specifically, $C_3$ plant biomass was significantly ($p < 0.01$) lower under warming than control, but no significant change was observed in $C_4$ and total plant biomass (Supplementary Fig. 1a), which results in a plant community shift towards relatively more $C_4$ plants. Although the statistical test is not significant, the gross primary production (GPP) was slightly increased by warming (Fig. 1c). Meanwhile, the net ecosystem exchange (NEE) was higher under warming than control due to lower ER, suggesting that the whole ecosystem acted as a C sink under the climate warming scenario (Fig. 1c). In addition, no overall differences were detected in total organic C (TOC), total nitrogen (TN) and soil pH (Supplementary Fig. 1b, c), but the amount of $NO_3^-$ was significantly ($p < 0.05$) higher under warming than control (Supplementary Fig. 1c). These alterations in ecosystem variables by warming are expected to lead to changes in soil respirations and microbial community functions.

**Temperature sensitivity of soil microbial respiration.** Soil surface $CO_2$ efflux was measured by using shallow (2–3 cm) PVC collars for $R_t$ and deep (70 cm) PVC tubes for $R_h$, with the differences between $R_t$ and $R_h$ calculated as $R_a$ (Supplementary Fig. 2 and Methods). Warming significantly ($p < 0.01$) stimulated $R_h$ by

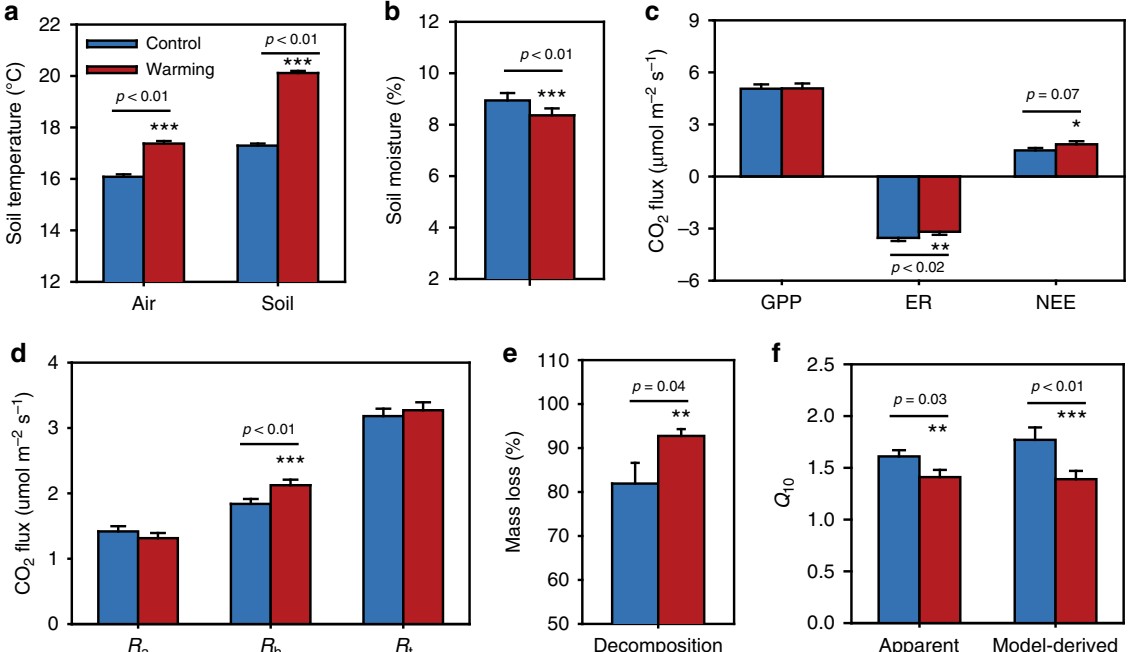

**Fig. 1 Warming effects on soil variables and ecosystem C fluxes. a** Air and soil surface (7.5 cm) temperatures averaged from 2010 to 2016. **b** Soil moisture averaged from 2010 to 2016. **c** Ecosystem C fluxes, which were estimated on the basis of the C amount from $CO_2$ emissions averaged from 2010 to 2016. GPP, gross primary productivity; ER, ecosystem respiration; NEE, net ecosystem C exchange. Positive values indicate C sink, and negative values represent C source. **d** in situ soil respirations averaged from 2010 to 2016. $R_a$, autotrophic respiration; $R_h$, heterotrophic respiration; $R_t$, soil total respiration. **e** Decomposition rate of standard cellulose filter paper (mass loss) in the field determined in 2016. **f** Apparent and model-derived temperature sensitivity ($Q_{10}$) of heterotrophic respiration ($R_h$) averaged from 2010 to 2016. Apparent $Q_{10}$ is estimated by fitting the curve of $R_h$ versus soil temperature based on the $Q_{10}$ method. Model-derived $Q_{10}$ is derived by calibrating the MEND model. Error bars represent standard errors of the means ($n = 4$ field plots examined over seven repeated measures from 2010 to 2016). The differences between warming and control were tested by the two-sided repeated measures ANOVA, indicated by *** when $p < 0.01$, ** when $p < 0.05$, * when $p < 0.10$. Source data are provided as a Source Data file.

8.0–28.1% across all years, which is consistent with results from a filter paper decomposition experiment that showed significantly ($p < 0.01$) higher decomposition rates under warming (Fig. 1e). However, warming appeared to suppress $R_a$, although it was not statistically significant (Fig. 1d), which may result from the decreased root activities along warming-induced plant community shift[7]. More than half of $R_t$ (58 and 65% for the control and warming plots) was from heterotrophic respiration, indicating that soil microbial community greatly contribute to soil $CO_2$ efflux[14]. Due to the opposing responses of $R_a$ and $R_h$ to warming, $R_t$ exhibited no significant change by warming across all years (Fig. 1d). Since our main interest is the response of microbial litter and SOM decomposition to warming, we primarily focused on $R_h$ for the majority of the following analyses. Notably, the root exclusion method by deep PVC tubes for partitioning $R_a$ and $R_h$ had some potential artifacts, including soil moisture and temperature changes, exclusions of plant detritus inputs, and soil microbial community changes[7], although these artifacts may be less problematic when we focused on the relative changes between treatments and controls. Soil moisture and temperature in the deep PVC tubes were not measured in this study, but previous study indicated that the similar root exclusion method (trenching method) can artificially increase soil moisture and temperature[26], and thus could overestimate soil $R_h$. Soil microbial community structure and biomass may not be significantly changed by root exclusion, as revealed by a previous study[27]. The severing roots by inserting PVC tube in soil may result in a transient increase of soil respiration[28]. In this study, soil $R_h$ was first measured at least 8 months after the insertion of PVC tube into soil, so the effects of decomposition of severing roots on measured $R_h$ should be minimized. However, it is highly possible that the

exclusion of root inputs to soil as dead roots and root exudates for a long time could underestimate soil $R_h$, and in turn overestimate soil $R_a$.

A wide range of different models have been developed to express the temperature sensitivity of SOM decomposition and respiration processes[29]. While many models are based on the exponential function characterized by the $Q_{10}$ or activation energy[4], the square root relationship[30] and the macromolecular rate theory (MMRT) equation[31] have also been proposed to enable the comparison of temperature sensitivity of microbial activity between habitats or organisms. The square root equation includes a theoretical minimum temperature for growth and activity, which allows one to more accurately estimate $Q_{10}$ below optimum temperature[30]. The core concept of the MMRT equation is that there exists an optimum temperature for enzyme and microbial activity[31], which overcomes the limit of temperature range for the applicability of the Arrhenius (including the $Q_{10}$ approach) and the square root equations. The optimum temperature in the MMRT equation could be ca. 30 °C[31,32] and 57–71 °C[33], which is generally above the temperature range of 0–40 °C for the validity of the Arrhenius relationship[12]. Given that 95% of the soil temperatures were below 30 °C in our study site and the $Q_{10}$ method has been widely accepted to interpret the temperature sensitivity in the biological and environmental research including most of the ESMs models[4,29], we adopted the $Q_{10}$ approach (see Methods) to examine the apparent temperature sensitivity of microbial respiration (>7 years) and their underlying mechanisms. This also allows us to directly compare our results to the vast amount of existing studies and interpret the temperature sensitivity in a generally accepted framework.

Significant ($p < 0.05$) or marginally significant ($p < 0.10$) apparent $Q_{10}$ estimates were observed under both control and warming treatments in all years except 2011 (Supplementary Fig. 3). Therefore, the apparent relationship between $R_h$ and soil temperature follow a monotonic exponential equation in most years. The poor fit of apparent $Q_{10}$ in 2011 and 2012 is most likely due to the suppression rather than enhancement of microbial respiration under warming, which could be explained by the higher temperatures (e.g., >30 °C) beyond the optimal temperature for microbial respiration[32] and/or the confounding effects of environmental factors other than temperature (e.g., soil moisture)[4]. In average, the apparent $Q_{10}$ estimates were significantly ($p = 0.03$) higher under control ($1.61 \pm 0.06$) than warming ($1.41 \pm 0.07$), suggesting a $12.0 \pm 3.7\%$ decrease in the temperature sensitivity of soil $R_h$ across 7 years of warming (Fig. 1f). However, the apparent temperature sensitivity estimate based on the field measurements are influenced by various other factors beyond temperature, including soil moisture, plants-derived substrate quality and availability, nutrient limitation influencing microbial enzyme production, experimental duration, and/or spatial heterogeneity, as well as uncertainty in instrumental measurements[4,8].

To further delineate the temperature sensitivity of SOM decomposition, ecosystem model-based inverse analysis was performed to untangle various complex soil processes[8,14,20] using the Microbial-ENzyme Decomposition (MEND) model (Supplementary Fig. 4a), which has been evaluated from laboratory to global scale[34–36]. Here we used the model-derived temperature sensitivity to distinguish the $Q_{10}$ estimated by process-based ecosystem modeling from the apparent $Q_{10}$ estimated by the relationship between respiration and temperature. The model-derived temperature sensitivity represents direct response of heterotrophic respiration to temperature change in the modeling context[37,38], as we used different response functions in MEND to represent the direct effects of soil pH, temperature, and moisture on various transformation processes[35]. By fitting all 7-year respiration data together, the model-derived $Q_{10}$ under warming was $1.39 \pm 0.09$, significantly lower ($p < 0.01$) than that under control ($1.77 \pm 0.12$) (Fig. 1f). The model-derived $Q_{10}$ values from our model-data fusion approach were comparable with the measured apparent $Q_{10}$ under both control and warming. Altogether, the above results indicate that there was a strong and persistent decrease in model-derived temperature sensitivity of microbial heterotrophic respiration under warming over the last 7 years.

**Mechanisms of soil microbial respiration**. The persistent decrease in temperature sensitivity of soil microbial respiration across different years under warming could be due to substrate depletion under warming. It has been argued that soil labile C becomes depleted by increased respiration in response to warming, which leads to a subsequent reduction in the rate of soil respiration[10]. In this study, several lines of evidence suggest that the decreased temperature sensitivity of microbial respiration was not mainly due to warming-induced substrate depletion. First, available C substrates are a tiny fraction of total soil C stocks and have rapid turnovers, but our BIOLOG results revealed that, after 7 years of warming, microbial metabolism underpinning the utilization ability of most available C substrates were considerably higher under warming than control (Supplementary Fig. 5). A reasonable explanation for the result is that soil C stocks, especially labile C pools as the sources of available C substrates[39,40] were relatively stable without substantial reduction, and can provide equal or more available C substrates after long-term warming, compared to the controls. Second, NEE was higher under warming than control (Fig. 1c), suggesting that more soil labile and/or recalcitrant C input as plant litter, root biomass or exudates counteracted the consumption of soil available C substrates by microbial respiration. Third, the unchanged annual soil C from 2010 to 2016 (Supplementary Fig. 1c) indicated the stability of soil total C under the long-term warming, which does not support the expectation garnered from the substrate depletion hypothesis. Altogether, these results suggested that the turnover of soil labile C may be accelerated by warming, but warming did not lead to the depletion of soil labile C. Therefore, the reduced temperature sensitivity of soil respiration appears to be less likely due to warming-induced substrate depletion, although the effects of substrate depletion could not be completely ruled out.

Warming-induced adaptive changes in microbial community composition and functional structure could also lead to the reduced temperature sensitivity of microbial respiration. To test this hypothesis, soil microbial communities of individual samples from 2010 to 2016 were all analyzed with deep amplicon sequencing of the 16S rRNA gene for bacteria and archaea, and the ITS for fungi, metagenomic shotgun sequencing, and functional gene arrays (GeoChip 5.0; Supplementary Table 1). Permutational multivariate analysis revealed that experimental warming significantly ($p < 0.03$) shifted microbial community taxonomic and functional structure (Table 1). These shifts were tightly linked to environmental factors as revealed by the Mantel test (Fig. 2a and Supplementary Fig. 6) and canonical correspondence analyses (CCA) (Supplementary Fig. 7). Interestingly, considerably less unexplained community variations were obtained based on GeoChip data (59.2%) than 16S (73.0%), ITS (77.4%) and shotgun sequencing data (73.3%) (Supplementary Fig. 8), indicating that GeoChip-based detection could be more effective in indicating the community dynamics in response to the changes in plant diversity, soil conditions, and time.

Generally, temperature is a primary driver of biological processes and can impact soil microbial communities at different organizational levels, based on the metabolic theory of ecology

**Table 1 Significance tests of the effects of warming and time on microbial community structures with permutational multivariate analysis of variance.**

| Effects | 16 S | | ITS | | GeoChip | | Metagenomic sequencing | | Metagenome EcoFUN-MAP | |
|---|---|---|---|---|---|---|---|---|---|---|
| | F | P | F | P | F | P | F | P | F | P |
| Warming (W) | 4.200 | 0.001 | 2.314 | 0.001 | 2.505 | 0.026 | 8.059 | 0.001 | 2.924 | 0.001 |
| Year (Y) | 2.432 | 0.001 | 1.595 | 0.001 | 12.216 | 0.001 | 4.398 | 0.001 | 2.323 | 0.001 |
| W × Y | 1.178 | 0.092 | 1.055 | 0.224 | 1.385 | 0.092 | 1.350 | 0.170 | 1.135 | 0.084 |

Permutational multivariate analysis of variance (Adonis) was used based on Bray–Curtis dissimilarity matrices. The two-way repeated measures ANOVA model was set as dissimilarity~warming × year + block using function adonis in R package vegan. The degree of freedom was 1 for warming treatment, 6 for year and 39 for residuals. Significant effects ($P \leq 0.05$) were shown in bold text. EcoFUN-MAP is a method designed for annotating metagenomic sequences by comparing them with functional genes used to fabricate GeoChip.

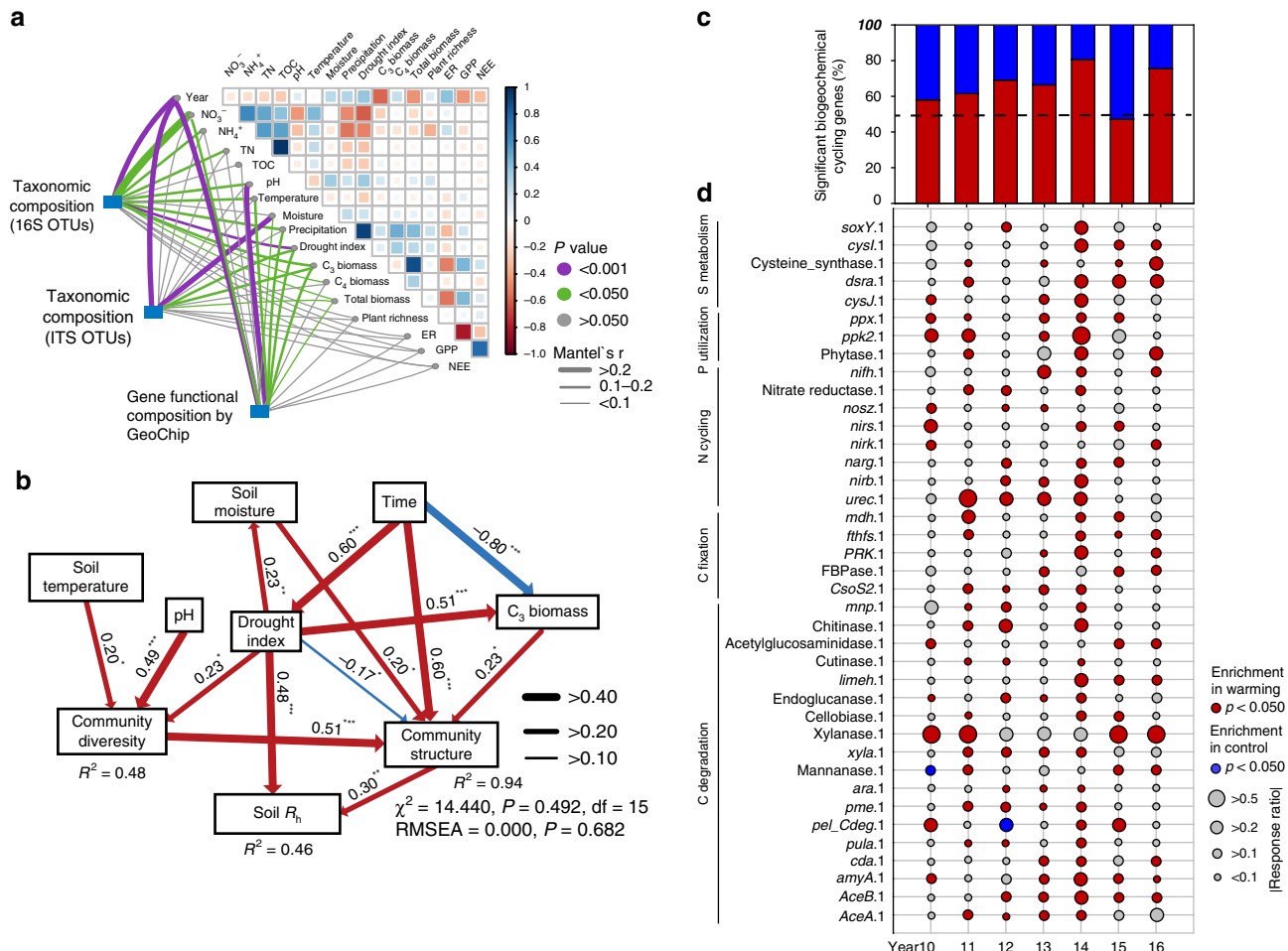

**Fig. 2 Feedback mechanisms of soil microbial communities to warming. a** Pairwise comparisons of environmental factors with a color gradient denoting Pearson's correlation coefficients. Taxonomic and functional community structures were related to each environmental factor by Mantel tests. Edge width corresponds to the Mantel's r statistic for the corresponding distance correlations, and edge color denotes the statistical significance. **b** The structural equation model (SEM) showing causal relationships among environmental factors, diversity and structure of microbial functional community, and soil $R_h$. Red and blue arrows represent significant positive and negative pathways, respectively. Arrow width is proportional to the strength of the relationship and bold numbers represent the standard path coefficients. The p values based on 1,000 bootstrapping for path coefficients are indicated by *** when $P <$ 0.001, ** when $P < 0.01$, * when $P < 0.05$. $R^2$ indicates the proportion of the variance explained for each dependent variable in the model. **c** The relative proportion of significantly warming-stimulated and significantly warming-inhibited genes in biogeochemical cyclings according to GeoChip data. Dash line represents that the abundance of warming-stimulated (red) genes are equal to the abundance of warming-inhibited (blue) genes. Significance is based on response ratio of each gene with 95% confidence intervals. Biogeochemical cycling genes included all genes involved in C degradation, C fixation, N cycling, phosphorus (P) utilization and sulfur (S) metabolism. **d** Bubble plot illustrating the enrichment of key biogeochemical cycling genes under warming (W) and control (C) treatments according to GeoChip data. Bubble color represents the significance (p-value) of gene enrichment based on response ratios. Bubble size represents the relative changes of gene enrichment based on response ratios. The biogeochemical cycling processes for these genes are shown in plot, and the full names of the genes in this plot are listed in Supplementary Table 4.

(MTE)[12,25,41]. The rising temperature under warming could act as a deterministic filtering factor to impose environmental selection on microorganisms, which can lead to significant shifts of soil microbial communities[25]. Consistent with these previous studies, soil temperature significantly ($p < 0.05$) correlated with the shifts of microbial composition and functional structure (Fig. 2a, Supplementary Fig. 6, 7). Furthermore, structural equation modeling (SEM)-based analysis indicated that soil temperature significantly influenced microbial functional structure through its effects on microbial diversity, further affecting soil $R_h$ (Fig. 2b). However, the shifts of microbial communities and soil $R_h$ may not be solely explained by the rising temperature under warming, since significant decreases of soil moisture were observed under warming, and strong correlations occurred between soil moisture and microbial composition and functional

structure (Figs. 1b, 2a). Previous studies provided clear evidences that soil moisture limitation can weaken the stimulation of warming on soil respiration[4,42]. Congruously, our SEM-based analysis suggested that soil moisture significantly ($p < 0.05$) affected soil $R_h$ through shifting microbial functional structure (Fig. 2b). It is highly possible that severe soil moisture limitation played more important role in changing soil microbial community and $R_h$ than temperature in the extremely drought year (2011), which led to no significant temperature sensitivities of soil microbial respiration observed in the year (Supplementary Fig. 3). In addition, warming can also alter soil microbial community structure indirectly through changing plant community structure, because the quantity and quality of soil C input from the plants differ depending upon the species[43]. In this study, $C_3$ plant biomass was significantly ($p < 0.05$) decreased by warming and

exhibited a direct effect on soil microbial function structure in the SEM-based analysis (Fig. 2b). All of these results indicated that the adaptive changes in microbial community composition and functional structure resulted from the combined effects of the increase of soil temperature, decrease of soil moisture, and changing plant community structure under long-term warming.

Warming-induced shifts of microbial functional diversity and structure led to significant changes of biogeochemical cycling processes, including C cycling (e.g., C degradation, C fixation) and nutrient-cycling processes (e.g., N fixation, denitrification, nitrification), phosphorus utilization and sulfur metabolism. Overall, the total abundance of biogeochemical cycling genes significantly ($p < 0.05$) stimulated by warming were considerably higher (58–80%) than those significantly inhibited by warming (20–42%) in all years except 2015 (Fig. 2c), although the interannual variations of environmental factors greatly influenced the composition of biogeochemical cycling genes. Similar pattern was also observed in microbial functional genes involved in C degradation, including those important for degrading starch (e.g., *amyA* encoding α-amylase), hemicellulose (e.g., *ara* encoding arabinofuranosidase), cellulose (e.g., cellobiase), chitin (e.g., chitinase) and vanillin/lignin (e.g., *mnp* encoding manganese peroxidase) (Supplementary Figs. 9a, 10). More specifically, larger numbers of individual genes involved in degrading various soil organic C were significantly increased by warming (95% confidence interval; Fig. 2d and Supplementary Fig. 9a) in most of the years, despite that warming effects on these C-degrading genes substantially changed across different years. The significant enrichment of C-degrading genes under warming may potentially enhance soil C degradation. In addition, the total abundances of warming-stimulated genes involved in N cycling (e.g., N fixation, denitrification, and nitrification), phosphorus utilization, and sulfur metabolism were higher than those of warming-inhibited genes in most of the years (Fig. 2d and Supplementary Fig. 9b–d), suggesting that the rates of nutrient-cycling processes could be stimulated by warming. Further analyses by CCA and Mantel test revealed that most of the genes important to C degradation and nutrient cycling had strong correlations to the $R_h$, $R_t$, and $Q_{10}$ (Supplementary Table 2 and 3), indicating that these functional genes are important in controlling the dynamics of soil respirations. In general, GeoChip hybridization data exhibited stronger correlations to various functional parameters than shotgun sequencing data, particularly for the heterotrophic $Q_{10}$ (Supplementary Tables 2, 3). All the above results indicate that the changes of microbial community composition and function are crucial for the reduced temperature sensitivity of soil $R_h$ under long-term experimental warming.

**Incorporating functional genes into ecosystem models.** Due to the importance of microbes in controlling soil $R_h$, as an exploratory effort, we further attempted to incorporate omics data into ecosystem models. Since traditional ecosystem models do not explicitly represent most microbial processes[44], the MEND model was employed, which explicitly represents microbial physiology and SOM decomposition catalyzed by oxidative or hydrolytic enzymes[36]. Because MEND model requires absolute quantitative information on hydrolytic and oxidative enzymes for SOM decomposition[36], GeoChip hybridization-based data were used, which is more effective to catch the community dynamic changes (Supplementary Fig. 8) as illustrated above.

The MEND model was calibrated with or without functional gene information. We referred the former to as gene-informed MEND (gMEND) and the latter as traditional MEND (tMEND). We constrained gMEND by achieving the highest correlation between MEND-modeled mean annual enzyme concentrations

and GeoChip-detected annual oxidative and hydrolytic gene abundances in addition to a best fit between observed and simulated $R_h$. Our results showed high correlations ($r = 0.74$ and 0.81 for oxidative and hydrolytic enzymes, respectively) between simulated enzyme concentrations and GeoChip-detected gene abundances (Supplementary Fig. 11a, b) in the control plots. Also, relatively low Mean Absolute Relative Errors (MARE = 14 and 22%, Supplementary Fig. 11c, d) were achieved between simulated and expected enzyme concentrations under warming conditions, which were the product of simulated enzyme concentrations under control and the warming-to-control ratio of GeoChip-detected gene abundances. The above modeling results indicated good agreements on the 7-year interannual variabilities between simulated enzyme concentrations and GeoChip-detected gene abundances.

As the MEND model uses different response functions to represent the effects of soil pH, temperature, and moisture on various transformation processes (Supplementary Table 5), the MEND model attempts to derive a $Q_{10}$ that specifically reflects the microbial and enzymatic responses to temperature change. This means that the direct effect of soil temperature may be distinguished from the effects of other environmental factors given the current model structure. To demonstrate the differentiation of the effects of soil temperature from moisture, we used gMEND to estimate the $R_h$ response to a single-factor change in soil temperature or moisture during the 7 year's experimental period. Compared to the MEND-simulated mean $R_h$ under control, changing soil temperature under warming would result in a 22.2% increase in $R_h$, whereas changing soil moisture would cause a decrease in $R_h$ by 8.1% (Supplementary Fig. 12). Therefore, both temperature and moisture effects greatly contribute the MEND-derived thermal adaptation effect, as both of them were taken into account in MEND simulations.

To test whether the inclusion of gene abundance data could reduce model uncertainty, we further quantified the uncertainty in parameters. Almost all of the 11 model parameters were better constrained by gMEND than by tMEND (Fig. 3a and Supplementary Fig. 13). The average coefficient of variation (CV) of model parameters was significantly reduced from 77% (tMEND) to 22% (gMEND) under control and from 39% (tMEND) to 17% (gMEND) under warming. Also, the MEND-simulated $R_h$ agreed well with the observed $R_h$ under warming and control (Fig. 3b: $R^2 = 0.53$ and 0.63, respectively). Compared to non-microbial terrestrial ecosystem model (TECO)[45], the MEND model improved $CO_2$ efflux fitting by 5% under control and by 19% under warming (Supplementary Fig. 14). We calibrated 10 parameters for TECO and 11 parameters for tMEND and gMEND, the Akaike information criterions (AIC) of the MEND models (−14.55 for warming and −38.30 for control) were smaller than those of the TECO model (−4.14 for warming and −34.79 for control), suggesting a better fit by the MEND model. In addition, the MEND-derived $Q_{10}$ was used to explore how much C loss is reduced by the thermal adaptation of soil microbial respiration ($Q_{10}$) under warming. Our results showed that the thermal adaptation of microbial respiration in the warming plots would reduce $11.6 \pm 7.5\%$ soil $R_h$, and thus reduce soil C loss, during the 7-year experimental period, compared to the scenarios without microbial thermal adaptation (Fig. 4a, b). This evidence for thermal adaptation in the present study contrasts with a recent meta-analysis of soil warming experiments, which found few significant differences in the temperature sensitivity of soil respiration between control and warmed plots across biomes and only limited evidence of acclimation of soil respiration to experimental warming[10]. This area of research clearly warrants additional study to understand differences in reported results among studies.

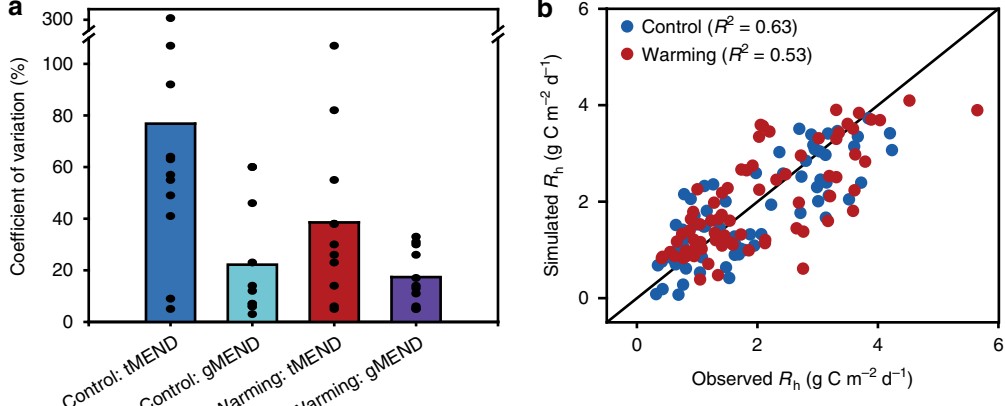

**Fig. 3 Model parameter uncertainty and modeling performance. a** The MEND model parameter uncertainty quantified by the Coefficient of Variation (CV). The bars show the mean CV values of the 11 parameters (See Supplementary Fig. 12 and Supplementary Table 8 for detailed description). The dots along each bar show the CV for each parameter. The tMEND refers to the traditional MEND model parameterization without gene abundances data. The gMEND denotes the improved MEND parameterization with gene abundances. **b** Comparison between gMEND-simulated and observed heterotrophic respiration ($R_h$) under control and warming ($R^2$ denotes the coefficient of determination).

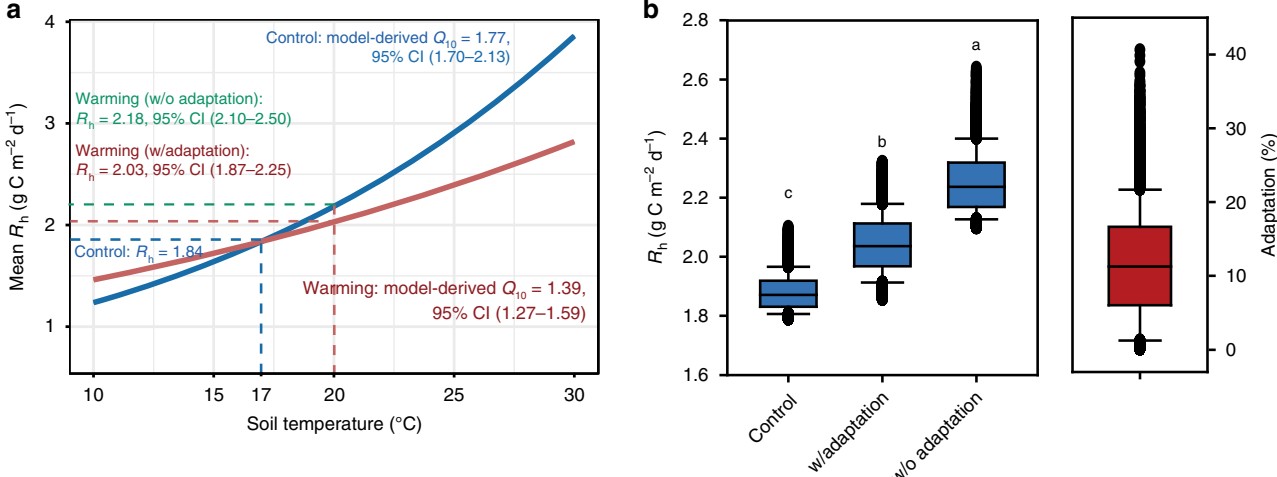

**Fig. 4 Microbial thermal adaptation of heterotrophic respiration ($R_h$) based on model-derived $Q_{10}$. a** Demonstration of $R_h$ thermal adaptation based on the mean $Q_{10}$ values. The mean annual soil temperature ($T$) during 2010–2016 was 17 °C and 20 °C under control and warming, respectively. The average model-derived $Q_{10} = 1.77$ under control and 1.39 under warming. The mean baseline $R_h = 1.84$ g C $m^{-2}$ $d^{-1}$ under control ($T = 17$ °C). The average $R_h = 2.03$ and 2.18 g C $m^{-2}$ $d^{-1}$ under warming ($T = 20$ °C) when thermal adaptation is considered (w/ Adaptation) or not considered (w/o Adaptation). The %Δ$R_h = 8.2$% means a 8.2% reduction in $R_h$ relative to the baseline $R_h = 1.84$ g C $m^{-2}$ $d^{-1}$ under control. 95% CI denotes the 95% confidence interval. **b** Thermal adaptation in $R_h$ when the uncertainties in model-derived $Q_{10}$ are considered. The mean thermal adaption (%Δ$R_h$) is 11.6%, which is different from the specific %Δ$R_h = 8.2$% derived from mean $Q_{10}$ values. However, both of these %Δ$R_h$ values are within the uncertainty range of %Δ$R_h$. Different letters for $R_h$ indicate significantly differences between the scenarios based on the Kruskal–Wallis test at a significance level of 0.05. The thermal adaptation (%) is quantified by the difference in $R_h$ between warming w/o Adaptation and w/ Adaptation as a percentage of the baseline $R_h$ under control (see "Methods" Eq. 9).

It should be noted that the model-derived $Q_{10}$ values may not represent the true intrinsic temperature sensitivity of microbial and enzyme activities. In this study, the MEND-derived $Q_{10}$ values were confined from 1.20 to 2.42 (tMEND) to a narrow range of 1.27–2.13 (gMEND), corroborating that $Q_{10}$ values of 2 or below are usually used in global C cycle modeling[29,38]. The MEND-derived $Q_{10}$ values ($1.77 \pm 0.12$ for control, and $1.39 \pm 0.09$ for warming) were close to those estimated from the TECO model in the current study ($1.79 \pm 0.09$ for control, and $1.50 \pm 0.15$ for warming), as well as a previous study with the TECO model (1.4–2.5)[37]. Our model-derived $Q_{10}$ under warming was similar to the detrended temperature sensitivity ($1.4 \pm 0.1$) estimated across 60 FLUXNET sites[38]. We also compared our

results to a meta-analysis of activation energy ($E_a$) and $Q_{10}$ values for cellulases and ligninases from ca. 60 publications (see Supplementary Table 11 and Supplementary Fig. 15). The MEND-derived mean $Q_{10}$ values under the control treatment were slightly (e.g., 2.7%) lower than the mean $Q_{10}$ of those studies (1.77 vs. 1.82). However, the model-derived mean $Q_{10}$ value under warming was at the lower bound of one standard deviation (1.39 vs. 1.38). We acknowledge that most C-degrading enzymatic processes have an activation energy of about 50–60 kJ $mol^{-1}$ (roughly equivalent to a $Q_{10}$ of 2.0–2.5)[4]. Therefore, the model-derived $Q_{10}$ values may fail to catch the intrinsic temperature sensitivity of microbial and enzyme activities, although we have attempted to separate the effects of temperature

from those of other potential confounding factors (e.g., soil moisture) through the process-based modeling. There are still other factors at the ecosystem level that likely limited the expression of the intrinsic temperature sensitivity of enzyme activity[4], which needs further research in future studies. In addition, limitations and uncertainties in model structure and parameterization could further hinder a thorough differential representation of the effects of multiple confounding factors (e.g., soil temperature and moisture, substrate supply and litter quality)[46] on enzyme activities and microbial carbon use efficiency (CUE), though our results showed no significant correlation between $Q_{10}$ and the temperature sensitivity of CUE (Supplementary Fig. 16). Despite that more effort should be devoted to improving the representation of multi-factor effects on soil respiration processes as well as confining the uncertainties in model structure, parameterization, and input data, microbially-enabled ecosystem modeling renders a significant advance in our understanding of microbial responses to the changes in temperature.

Through field measurements and process model-based simulations, our results demonstrated that thermal adaptation of microbial respiration persisted over the last 7 years, which is consistent with a recent long-term study on a forest ecosystem[15]. This study provides explicit, robust evidence of the persistence of thermal adaptation of microbial respiration to warming treatments and associated decreases in soil moisture over long periods. If this phenomenon holds over larger spatial scales across different ecosystems, thermal adaptation of soil microbial respiration globally may have a greater mitigating impact than expected on climate warming-induced $CO_2$ losses[47]. Our study also reveals that warming-induced thermal adaptation of soil respiration is significantly correlated with the adaptive changes in microbial community functional structure, which could dampen the potential positive C-climate feedbacks by reducing considerable amount of warming-induced heterotrophic respiration. In addition, although incorporating complex microbial information into global change models is extremely challenging[22], by parameterizing the microbial model with omics-based functional gene information, the uncertainty of key model parameters in MEND was substantially decreased, and its performance was considerably improved compared to non-microbial model. Thus, it is possible to improve the model predictive ability for projecting future environmental changes via a better representation of multi-factor effects on soil biogeochemical processes and a comprehensive assessment of microbial omics-based functional capacities. However, to generalize whether these microbial mechanisms and metagenomics-enabled modeling strategy obtained in this grassland ecosystem are applicable to other ecosystems requires further long-term studies under realistic field settings.

## Methods
**Site description and sampling**. This experimental site was established in July 2009 at the Kessler Atmospheric and Ecological Field Station (KAEFS) in the US Great Plains in McClain County, Oklahoma (34°59′N, 97°31′W)[14,48]. Experimental design and site description were described in detail previously[25]. Briefly, *Ambrosia trifida*, *Solanum carolinense* and *Euphorbia dentate* belonging to $C_3$ forbs, and *Tridens flavus*, *Sporobolus compositus* and *Sorghum halapense* belonging to $C_4$ grasses are dominant in the site[25,48]. Annual mean temperature is 16.3 °C and annual precipitation is 914 mm, based on Oklahoma Climatological Survey data from 1948 to 1999. The soil type of this site is Port–Pulaski–Keokuk complex with 51% of sand, 35% of silt and 13% of clay, which is a well-drained soil that is formed in loamy sediment on flood plains. The soil has a high available water holding capacity (37%), neutral pH and 1.2 g cm$^{-3}$ bulk density with 1.9% total organic matter and 0.1% total nitrogen (N)[25,48]. Four blocks were used in the field site experiment, in which warming is a primary factor. Two levels of warming (ambient and +3 °C) were set for four pairs of 2.5 m × 1.75 m plots by utilizing a real or dummy infrared radiator (Kalglo Electronics, Bethlehem, PA, USA). In the warmed plots, a real infrared radiator was suspended 1.5 m above the ground, and the dummy infrared

radiator was suspended to simulate a shading effect of the device in the control plots.

In this study, eight surface (0–15 cm) soil samples, four from the warmed and four from the control plots, were collected annually at approximately the date of peak plant biomass (September or October) from 2010 to 2016. Three soil cores (2.5 cm diameter × 15 cm depth) were taken by using a soil sampler pipe in each plot and composited to have enough samples for soil chemistry, microbiology and molecular biology analyses. A total of 56 soil samples were analyzed in this study.

**Environmental and soil chemical measurements**. Precipitation data were obtained from the Oklahoma Mesonet Station (Washington Station)[48] located 200 m away from our experiment site, and 12-month version of the standardized precipitation-evapotranspiration index (SPEI-12) was used as annual drought index[49]. Air temperature, soil temperature and volumetric soil water content were described in detail previously[25]. Specifically, air temperature and soil temperature at the depth of 7.5 cm in the center of each field plot were measured by using Constantan-copper thermocouples wired to a Campbell Scientific CR10x data logger (Campbell Scientific, UT, USA). A portable time domain reflectometer (Soil Moisture Equipment Corp.) was used to measure soil moisture from the soil surface to a 15-cm depth once or twice a month. Three measurements of soil moisture were performed in each plot and the average of three technical replicates were used in further analyses.

All soil samples were analyzed to determine soil total organic carbon (TOC), total nitrogen (TN), soil nitrate ($NO_3^-$) and ammonia ($NH_4^+$) by the Soil, Water, and Forage Analytical Laboratory at Oklahoma State University (Stillwater, OK, USA). Soil pH was measured using a pH meter with a calibrated combined glass electrode[50].

**Aboveground plant communities**. Aboveground plant community investigations were annually conducted at peak biomass (usually September)[48,51]. Aboveground plant biomass, separated into $C_3$ and $C_4$ species, was indirectly estimated by a modified pin-touch method[48,51]. Detailed description of biomass estimation is provided by Sherry et al.[52]. A pin frame used in this study is 1 m long and have 10 pins 10 cm apart at 30° from vertical. Pins with a 0.75 m length were raised within the frame to count hits up to 1 m high (hits over 1 m are negligible at this site). The pin frame was placed in the center of each plot to record the contact numbers of the pins separately with $C_3$ and $C_4$ plants (e.g., leaves and stems). The contact numbers of $C_3$ and $C_4$ plants were then used to estimate plant biomass using calibration equations derived from calibration plots, which were located near the experimental plots. Biomass in the calibration plots was clipped at a height of 10 cm above the ground at approximately the date of peak plant biomass (September or October). All of the species in plant community within each plot were identified to estimate species richness. Clipped plant materials were oven-dried and then correlated with the total contact number. $C_3$ and $C_4$ plant biomasses were estimated by using the calibration equation of contact number and plant biomass. All of the species within each plot were identified to estimate species richness of plants.

**Ecosystem C fluxes and soil respiration**. Ecosystem C fluxes and soil respirations were measured once or twice a month between 10:00 and 15:00 (local time) from January 2010 to December 2016 by following previous methods[14,48]. One square aluminum frame (0.5 m × 0.5 m) was inserted in the soil at 2 cm depth in each plot to provide a flat base between the soil surface and the $CO_2$ sampling chamber. NEE and ecosystem respiration (ER) were measured using LI-6400 portable photosynthesis system (LI-COR). Gross primary productivity (GPP) was estimated as the difference between NEE and ER. Meanwhile, soil surface respiration was monthly measured using a LI-8100A soil flux system attached to a soil $CO_2$ flux chamber (LI-COR). Measurements were taken above a PVC collar (80 cm$^2$ in area and 5 cm in depth) and a PVC tube (80 cm$^2$ in area and 70 cm in depth) in each plot. The PVC tube was permanently fixed on the ground to cut off old plant roots and prevent new roots from growing inside the tube. Any aboveground parts of living plants were removed from the PVC tubes and collars before each measurement. The $CO_2$ efflux measured above the PVC tubes represented heterotrophic respiration ($R_h$) from soil microbes, while that measured above the PVC collars represented soil total respiration ($R_t$) including heterotrophic and autotrophic respiration ($R_h$ and $R_a$) from soil microbes and plant root, respectively.

**Soil decomposition rate**. Weighted cellulose filter paper (Whatman CAT No. 1442-090) was placed into fiberglass mesh bags and placed vertically at 0–10 cm soil depth in each plot in March 2016. All of decomposition bags were collected back in September 2016, rinsed and dried at 60 °C for weighing. The percentage of mass loss was calculated to represent soil decomposition rate.

**Molecular analyses of soil samples**. The C substrate utilization patterns of soil microbial communities in 2016 were analyzed by BIOLOG EcoPlate$^{TM}$ (BIOLOG). The BIOLOG EcoPlate$^{TM}$ contains 31 of the most useful labile carbon sources for soil community analysis, which are repeated three times in each plate. In this study, the plates with diluted soil supernatant (0.5 g soil with 45 mL 0.85% NaCl) were incubated in a BIOLOG OmniLog PM System at 25 °C for 4.5 days. The color change of each well was shown as absorbance curve. The net area under the

 

absorbance versus time curve was calculated to represent physiological activity of various C sources[53]. The average value from three replicates was used for analyses in this study.

Soil total DNA was extracted from 1.5 g soil by freeze-grinding and SDS-based lysis[54], and purified with a MoBio PowerSoil DNA isolation kit (MoBio Laboratories)[25]. Then, 10 ng DNA per sample were used for library construction and amplicon sequencing. Amplicons sequencing was performed with cautions in terms of experimental preparations and data analyses to ensure sequence representativeness and semi-quantitative nature[55]. The V4 region of bacterial and archaeal 16S rRNA genes were amplified with the primer set 515F (5′-GTGCC AGCMGCCGCGGTAA-3′) and 806R (5′-GGACTACHVGGGTWTCTAAT-3′), and fungal ITSs between 5.8S and 28S rRNA genes were amplified with the primer set ITS7F (5′-GTGARTCATCGARTCTTTG-3′) and ITS4R (5′-TCCTC CGCTTATTGATATGC-3′). PCR products from different samples were sequenced on a MiSeq platform (Illumina, Inc.) using 2 × 250 pair-end sequencing kit. Raw sequences were submitted to our Galaxy sequence analysis pipeline (http://zhoulab5.rccc.ou.edu:8080) to further analyze according to the protocol in the pipeline[25]. Finally, OTUs were clustered by UPARSE[56] at 97% identity for both 16S rRNA gene and ITS. All sequences were randomly resampled to 30,000 sequences for 16S rRNA gene and 10,000 sequences for ITS per sample. Representative sequences of OTUs were annotated taxonomically by the Ribosomal Database Project (RDP) Classifier with 50% confidence estimates.

GeoChip 5.0 M, a functional gene array[57], was used for all 56 samples from 2010 to 2016. GeoChip hybridization, scanning and data processing were performed in the Institute for Environmental Genomics, University of Oklahoma[57,58]. Specifically, 800 ng of purified soil DNA of each sample was mixed with 5.5 μl random primers (Life Technologies, random hexamers, 3 μg/μl), diluted with nuclease-free water to 35 μl, heated to 99 °C for 5 min, and placed on ice immediately. The labeling master mix (15 μl), including 0.5 μl of Cy-3 dUTP (25 nM; GE Healthcare), 2.5 μl of dNTP (2.5 mM dTTP, 5 mM dAGC-TP), 1 μl of Klenow (imer; San Diego, CA; 40 U ml⁻¹), 5 μl Klenow buffer, and 2.5 μl of water, was added in the sample mixed solution. The samples were incubated at 37 °C for 6 h in a thermocycler, and then incubated at 95 °C for 3 min to inactivate the enzyme. Subsequently, samples were protected from the light as much as possible. Labeled DNA was cleaned using a QIAquick purification kit (Qiagen) according the manufacturer's instructions and then dried thoroughly in a SpeedVac (45 °C, 45 min; ThermoSavant).

Labeled DNA was resuspended into 27.5 μl of DNase-free water, and then mixed completely with 99.4 μl of hybridization solution, containing 63.5 μl of formamide (10% final concentration), 2 × HI-RPM hybridization buffer, 12.7 μl of 10 × aCGH blocking agent, 0.05 μg/μl Cot-1 DNA, and 10 pM CORS[58]. The mixed solution was denatured at 95 °C for 3 min, and then incubated at 37 °C for 30 min. The DNA solution was centrifuged at 6000 × g for 1 min to collect liquid at the bottom of the tube. 110 μl of the solution was pipetted into the center of the well of the gasket slide. The array slide was placed on the gasket slide, sealed using a SureHyb chamber, hybridized at 67 °C for 24 h at 20 rpm in a hybridization oven. After hybridization, slides were washed in room temperature with Wash Buffer 1 (Agilent) and Wash Buffer 2 (Agilent).

The slides were imaged as a Multi-TIFF with a NimbleGen MS200 Microarray Scanner (Roche NimbleGen, Inc., Madison, WI, United States). The raw signals from NimbleGen were submitted to the Microarray Data Manager on our website (http://ieg.ou.edu/microarray), cleaned, normalized and analyzed using the data-analysis pipeline. Briefly, probe quality was assessed, and poor or low signal probes were removed. Probe spots with coefficient of variance (CV; probe signal SD/signal) >0.8 were removed. Then, the signal-to-noise ratio (SNR) was calculated. As suggested by Agilent, the average signal of Agilent's negative control probes within each subarray was used as the background signal for the probes in that subarray instead of the local background typically used. The signal intensity for each spot was corrected by subtracting the background signal intensity. If the net difference was<0, the spots were excluded from subsequent analysis[57]. The average signal intensity of CORS was calculated for each subarray, and the maximum average value among all subarrays was used to normalize the signal intensity of samples in each array. Second, the sum of the signal intensity was calculated for each array, and the maximum sum value was used to normalize the signal intensity of all spots in each array, which produced a normalized value for each spot in each array.

Metagenomic library of all samples was prepared using a KAPA Hyper Prep Kit and sequenced at the Oklahoma Medical Research Foundation's Genomics Core using the Illumina HiSeq 3000 platform with a 2 × 150 bp paired-end kit. A total of 8.18 billion reads were obtained from all 56 samples, and 80 million reads were randomly resampled from each sample to perform data processing. Open reading frames (ORFs) were predicted on non-16S encoding reads using FragGeneScan with the 0.5% Illumina sequencing error model and the default settings. The predicted amino acid (a.a.) sequences for ORFs were then searched against the M5NR database using BLAST, with the following settings: a.a. identity >30%, aligned length >20 a.a., and e-value <1e−10. Read matching genes was incorporated in the SEED database. The numbers of annotated reads were taken as a proxy of abundance of the SEED subsystems[57]. Meanwhile, all reads were also submitted to our EcoFUN-MAP pipeline (http://www.ou.edu/ieg/tools/data-analysis-pipeline.html) to fish out shotgun sequence reads of important environmental functional genes used to fabricate GeoChip according to the protocol in the pipeline[59].

**Model simulations (TECO and MEND model).** Daily GPP values were obtained from a corrected 8-day GPP product based on the MODIS GPP (MOD17A2/MOD17A2H)[60]. We assign the same daily GPP values for the 8-day period. Meanwhile, datasets measured in both control and warmed plots across all years were also used for model simulations, including soil temperature and moisture, heterotrophic respiration, and the GeoChip-detected enzyme densities.

To examine temperature sensitivity of microbial heterotrophic respirations, the measured field $R_h$ in warmed and control plots was fitted with the exponential equation[4] (Eq. (1)) on yearly basis or across all years. In the equation, $R$ is $R_h$, $T$ is soil temperature, $R(T_{ref})$ is the respiration rate at the reference temperature ($T_{ref}$). The $Q_{10}$ estimated by the observed respiration data was called apparent $Q_{10}$ of respiration in this study.

$$R(T) = R(T_{ref}) \times Q_{10}^{(T-T_{ref})/10}. \quad (1)$$

In the MEND model, the parameter $Q_{10}$ is used to characterize the unconfounded temperature sensitivity of SOM decomposition and heterotrophic respiration. Constrained $Q_{10}$, which is the model-derived $Q_{10}$, is estimated by model fitting constrained by available observations including respiration and gene abundances, were obtained for the control and warming plots by incorporating respiration and microbial information into the MEND model parameterization process, which we called the model-derived $Q_{10}$ of soil respirations[38]. The model-derived $Q_{10}$ can represent the direct response to temperature versus the confounded effects of multiple factors, such as soil moisture and substrate availability.

The non-microbial terrestrial ecosystem (TECO) model is a variant of the CENTURY model[61] that is designed to simulate C input from photosynthesis, C transfer among plant and soil pools, and respiratory C releases to the atmosphere (Supplementary Fig. 4b). C dynamics in the TECO model can be described by a group of first-order ordinary differential equations, where the turnover rates are modified by soil temperature ($T$) and moisture ($W$)[45]. Prior ranges of turnover rates were based on Weng and Lu[62]. The prior ranges of $Q_{10}$ were based on the ranges of apparent $Q_{10}$ of $R_h$ per treatment[4]. We assumed that the parameters distributed uniformly in their prior ranges[8]. We used the Shuffled Complex Evolution (SCE) algorithm to determine model parameters[36]. We also applied the probabilistic inversion (Markov Chain Monte Carlo) to quantity parameter uncertainties[63]. By performing TECO modeling, daily heterotrophic respiration was simulated for both warmed and control plots from 2010 to 2016. The coefficient of determination ($R^2$) was used to estimate the model performance between observed and simulated respiration[64].

The Microbial-ENzyme Decomposition (MEND) model (Supplementary Fig. 4a) describes the SOM decomposition processes by explicitly representing relevant microbial and enzymatic physiology[36]. The SOM pool consists of two particulate organic matter (POM) pools and one mineral-associated organic matter (MOM) pool. The two POMs are decomposed by oxidative and hydrolytic enzymes, respectively. The MOM is decomposed by a generic enzyme group (EM). Model state variables, governing equations, component fluxes and parameters are described in Supplementary Table 6–9, respectively. A model parameter (reaction rate) in MEND may be modified by soil water potential, temperature, or pH[36]. MEND represents microbial dormancy, resuscitation, and mortality, and enzymatic decomposition in response to changes in moisture, as well as shifting of microbial and enzymatic activities with changing temperature[35]. The temperature response functions are described by the monotonic exponential equation (characterized by the activation energy) or the $Q_{10}$ method[65], where the $Q_{10}$ method was used in this study.

The model parameters are determined by achieving high goodness-of-fits of model simulations against experimental observations, such as heterotrophic respiration ($R_h$), microbial biomass carbon (MBC), gene abundances of oxidative (EnzCo) and hydrolytic enzymes (EnzCh) in this study (Supplementary Table 10). We implemented multi-objective calibration of the model[35]. Each objective evaluates the goodness-of-fit of a specific observed variable, e.g., $R_h$, MBC, or gene abundances (Supplementary Table 10). Note that the GeoChip gene abundances were used to constrain the MEND modeling as additional objective functions. The parameter optimization is to minimize the overall objective function ($J$) that is computed as the weighted average of multiple single-objectives (Supplementary Table 9)[36]

$$J = \sum_{i=1}^{m} w_i \cdot J_i, \quad (2)$$

$$\sum_{i=1}^{m} w_i = 1 \quad \text{with} \quad w_i \in [0, 1], \quad (3)$$

where $m$ denotes the number of objectives and $w_i$ is the weighting factor for the $i$th ($i = 1, 2, …, m$) objective ($J_i$). In this study, $J_i$ ($i = 1, 2, 3, 4$) refers to the objective function value for $R_h$, MBC, EnzCo, and EnzCh, respectively. Because we have far more $R_h$ observations (e.g., 74 in control or warmed cases) than the other variables and $R_h$ is the most important variable in soil C studies, we assign a much higher weighting factor to $R_h$ than the other three objective functions (MBC, EnzCo, and EnzCh), i.e, $w_1 = 5/8$ and $w_2 = w_3 = w_4 = 1/8$.

As the overall objective function $J$ is minimized in the parameter optimization process, the individual objective function $J_i$ may be calculated as $(1-R^2)$, $(1-r)$, or

 

MARE:

$$R^2 = 1 - \frac{\sum_{i=1}^{n}[Y_{sim}(i) - Y_{obs}(i)]^2}{\sum_{i=1}^{n}[Y_{obs}(i) - \bar{Y}_{obs}]^2}, \quad (4)$$

$$MARE = \frac{1}{n}\sum_{i=1}^{n}\left|\frac{Y_{sim}(i) - Y_{obs}(i)}{Y_{obs}(i)}\right|, \quad (5)$$

$$r = \frac{\sum_{i=1}^{n}[Y_{obs}(i) - \bar{Y}_{obs}]\cdot[Y_{sim}(i) - \bar{Y}_{sim}]}{\sqrt{\sum_{i=1}^{n}[Y_{obs}(i) - \bar{Y}_{obs}]^2}\cdot\sqrt{\sum_{i=1}^{n}[Y_{sim}(i) - \bar{Y}_{sim}]^2}}, \quad (6)$$

where $R^2$ denotes the Coefficient of Determination[36,66]. The $R^2$ quantifies the proportion of the variance in the response variables that is predictable from the independent variables. A higher $R^2$ ($R^2 \leq 1$) indicates better model performance. MARE is the Mean Absolute Relative Error (MARE) and lower MARE values (MARE $\geq 0$) are preferred[36,67]. MARE represents the averaged deviations of predictions ($Y_{sim}$) from their observations ($Y_{obs}$). $r$ is Pearson correlation coefficient and higher $r$ values ($|r| \leq 1$) means better model performance. $n$ is the number of data; $Y_{obs}$ and $Y_{sim}$ are observed and simulated values, respectively; and $\bar{Y}_{obs}$ and $\bar{Y}_{sim}$ are the mean value for $Y_{obs}$ and $Y_{sim}$, respectively.

Different objective functions are used to quantify the goodness-of-fit for different variables (Supplementary Table 9), depending on the measurement method and frequency of variables. The $R^2$ is used to evaluate the variables (e.g., soil respiration) that are frequently measured and the absolute values can be directly compared between observations and simulations. The MARE is used to evaluate the variables (e.g., microbial biomass and enzyme concentrations) with only a few measurements and the absolute values can be directly compared. When the absolute values cannot be directly compared, the correlation coefficient ($r$) between original or transformed (e.g., logarithmic transformed) observations and simulations will be used. For example, the gene abundances from metagenomics or GeoChip analysis cannot be directly compared to the enzyme concentrations or activities in the MEND model. However, we may assume correlation could be found between the measured and modeled values with a certain transformation or normalization.

We used the Shuffled Complex Evolution (SCE) algorithm to determine model parameters for the control soil and the warming soil respectively. SCE is a stochastic optimization method that includes competitive evolution of a "complex" of points spanning the parameter space and the shuffling of complexes[68].

The parameter uncertainty in the MEND model was quantified by the Critical Objective Function Index (COFI) method[36]. The COFI method is based on a global stochastic optimization technique (e.g., SCE in this study). It also accounts for model complexity (represented by the number of model parameters) and observational data availability (represented by the number of observations). The confidence region of parametric space were determined by selecting those parameter sets resulting in objective function values ($J$) less than the COFI value ($J_{cr}$) from the feasible parameter space[36].

To examine how much soil C loss is reduced by the soil microbial thermal adaptation under warming, we further calculated heterotrophic respiration ($R_h$) under warming without thermal adaptation (w/o Adaptation). That is, we estimated the mean $R_h$ changing with soil temperature that under warming, however, we kept the same range of $Q_{10}$ as that under control[13,15]. The $R_h$ changing with soil temperature is described by the $Q_{10}$ method similar to Eq. (1):

$$R_h(T) = R_h(T_{ref}) \times Q_{10}^{(T - T_{ref})/10}, \quad (7)$$

where $R_h(T)$ and $R_h(T_{ref})$ are the $R_h$ (g C m$^{-2}$ d$^{-1}$) at soil temperature ($T$) and reference temperature ($T_{ref}$), respectively; and $T_{ref} = 10\,°C$ in this study.

We quantified the thermal adaptation effect (w/Adaptation) by taking into account the uncertainties in model-derived $Q_{10}$ estimated by the MEND model. First we calculated the $R_h$ fluxes (g C m$^{-2}$ d$^{-1}$) at the mean annual soil temperature under control, e.g., $R_h^{CT}$ under $T = 17\,°C$ and $Q_{10} = 1.77$ with 95% confidence interval (CI) of 1.70–2.13. Second we calculated $R_h$ under warming with thermal adaptation ($R_h^{wA}$ under $T = 20\,°C$ and $Q_{10} = 1.39$ with 95% CI of 1.27–1.59) and $R_h$ under warming without thermal adaptation ($R_h^{woA}$ under $T = 20\,°C$ and $Q_{10} = 1.77$ with 95% CI of 1.70–2.13). We then calculated the reduction in $R_h$ due to thermal adaptation as

$$\Delta R_h^{woA-wA} = R_h^{woA} - R_h^{wA}. \quad (8)$$

Finally, we calculated the thermal adaptation effect as the percent reduction in $R_h$ due to thermal adaptation relative to the baseline $R_h$, i.e, the mean $R_h$ in the control plot ($R_h^{CT}$)

$$\%\Delta R_h = \Delta R_h^{woA-wA}/R_h^{CT} \times 100\%. \quad (9)$$

**Statistical analysis**. All statistical analyses were carried out using R software 3.1.1 with the package vegan[69] (v.2.3-5) and pgirmess[70] (v.1.5.8) unless otherwise indicated. The difference of various variables between warming and control was tested by repeated measures analysis of variance (ANOVA). The non-parametric multivariate analysis of variance (Adonis) were used to test the difference of microbial community taxonomic and functional structures considering the blocked

split-plot design[25]. CCA and Mantel test were performed to examine the linkage between environmental variables and microbial community structure/subcategories of functional genes. The significance of the CCA model was tested by analysis of variance (ANOVA). CCA-based variation partitioning analysis (VPA) was performed to evaluate how much different types of environmental variables influences microbial community phylogenetic and functional structures[14]. Structural equation model (SEM) was used to explore how warming-induced environmental variables affected soil microbial communities and heterotrophic respiration. Response ratio (RR) was used to compute the effects of warming on functional genes involved in C cycling and nutrient-cycling processes from GeoChip data using the formula RR = ln (warming/control)[59]. The non-parametric Kruskal–Wallis method[70] was used to test the significance of difference in model parameter values or the $R_h$ under different scenarios at a significance level of 0.05.

**Reporting summary**. Further information on research design is available in the Nature Research Reporting Summary linked to this article.

## Data availability
DNA sequences of 16S rRNA gene and ITS amplicons were available in NCBI Sequence Read Archive under project no. PRJNA331185. Raw shotgun metagenomic sequences are deposited in the European Nucleotide Archive (http://www.ebi.ac.uk/ena) under study no. PRJNA533082. GeoChip signal intensity data can be accessed through the URL (https://www.ou.edu/ieg/publications/datasets). All other relevant data are available in Supplementary Information. Source data are provided with this paper.

## Code availability
MEND model codes are accessible at https://github.com/wanggangsheng/MENDokw.git.

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

## Acknowledgements

The authors are grateful to the numerous former laboratory members for their help in maintaining the experimental site. We thank Drs. Xiangming Xiao, Xiaocui Wu, and Rajen Bajgain at University of Oklahoma for providing data support. This study is funded by the US Department of Energy, Office of Science, Genomic Science Program under Award Number DE-SC0004601 and DE-SC0010715, and the Office of the Vice President for Research at the University of Oklahoma. X.G., Q.G., and X.Z. acknowledge China Scholarship Council (CSC) for support.

## Author contributions

All authors contributed intellectual input and assistance to this study and manuscript preparation. Research questions and experimental strategy were developed by J.Z., E.A.G.S., Y.L., K.T.K., J.R.C., C.R.P., and J.M.T. Field management was carried out by J.F., M.Y., C.G.J., S.N., D.L., X.X., L.J., L.Y.W., A.Z., F.L., B.W., and J.D.V.N. Sampling collections, DNA preparation and MiSeq sequencing analysis were carried out by X.G., L.C. and X.Z. Geo-Chip hybridization and shotgun sequencing analysis were performed by X.G., X.Z., and R.T. Soil chemical and substrate analyses were carried out by X.Z., L.H., A.E., and L.W.W. Modeling was done by Q.G. and G.W. Various statistical analyses were carried by X.G., Z.S., and D.N. Assistance in data interpretation was provided by X.L., Y.Y., and Z.H. All data analysis and integration were guided by J.Z. The paper was written by J.Z., X.G., and Q.G. with help from G.W. Considering their contributions in terms of site management, data

collection, analyses and/or integration over the last 8 years, X.G., Q.G., M.Y., and G.W. are listed as co-first authors.

## Competing interests

The authors declare no competing interests.

## Additional information

Xue Guo [1,2,3,20], Qun Gao[1,2,3,20], Mengting Yuan [4,20], Gangsheng Wang [2,3,20], Xishu Zhou[2,3,5], Jiajie Feng[2,3], Zhou Shi[2,3], Lauren Hale [2,3], Linwei Wu [2,3], Aifen Zhou[2,3], Renmao Tian[2,3], Feifei Liu [2,3], Bo Wu[2,3,6], Lijun Chen[2], Chang Gyo Jung [7], Shuli Niu[8,9], Dejun Li[10,11], Xia Xu[12], Lifen Jiang[7], Arthur Escalas [2,3], Liyou Wu[2,3], Zhili He[2,3,6,13], Joy D. Van Nostrand [2,3], Daliang Ning [2,3], Xueduan Liu[5], Yunfeng Yang [1], Edward. A. G. Schuur [7], Konstantinos T. Konstantinidis[14], James R. Cole[15], C. Ryan Penton [16,17], Yiqi Luo [7], James M. Tiedje[15] & Jizhong Zhou [1,2,3,18,19✉]

[1]State Key Joint Laboratory of Environment Simulation and Pollution Control, School of Environment, Tsinghua University, Beijing, China. [2]Institute for Environmental Genomics, University of Oklahoma, Norman, Oklahoma, USA. [3]Department of Microbiology and Plant Biology, University of Oklahoma, Norman, Oklahoma, USA. [4]Department of Environmental Science, Policy, and Management, University of California, Berkeley, California, USA. [5]School of Minerals Processing and Bioengineering, Central South University, Changsha, Hunan, China. [6]Environmental Microbiomics Research Center and School of Environmental Science and Engineering, Sun Yat-sen University, Guangzhou, China. [7]Center for Ecosystem Science and Society, Department of Biological Sciences, Northern Arizona University, Flagstaff, Arizona, USA. [8]Institute of Geographic Sciences and Natural Resources Research, Chinese Academy of Sciences, Beijing, China. [9]University of Chinese Academy of Sciences, Beijing, China. [10]Key Laboratory of Agro-ecological Processes in Subtropical Region, Institute of Subtropical Agriculture, Chinese Academy of Sciences, Changsha, Hunan, China. [11]Huanjiang Observation and Research Station for Karst Ecosystem, Chinese Academy of Sciences, Huanjiang, Guangxi, China. [12]College of Biology and the Environment, Co-Innovation Center for Sustainable Forestry in Southern China, Nanjing Forestry University, Nanjing, China. [13]Southern Laboratory of Ocean Science and Engineering (Zhuhai), Zhuhai, China. [14]School of Civil and Environmental Engineering and School of Biological Science, Georgia Institute of Technology, Atlanta, Georgia, USA. [15]Center for Microbial Ecology, Michigan State University, East Lansing, Michigan, USA. [16]College of Letters and Sciences, Faculty of Science and Mathematics, Arizona State University, Mesa, AZ, USA. [17]Center for Fundamental and Applied Microbiomics, Biodesign Institute, Arizona State University, Tempe, AZ, USA. [18]School of Civil Engineering and Environmental Sciences, University of Oklahoma, Norman, Oklahoma, USA. [19]Earth and Environmental Sciences, Lawrence Berkeley National Laboratory, Berkeley, California, USA. [20]These authors contributed equally: Xue Guo, Qun Gao, Mengting Yuan, Gangsheng Wang. ✉email: jzhou@ou.edu

