## [Peer Review File · Nature Communications]

Editorial Note: This manuscript has been previously reviewed at another journal that is not operating a transparent peer review scheme. This document only contains reviewer comments and rebuttal letters for versions considered at *Nature Communications*. Mentions of the other journal have been redacted.

Reviewers' comments:

Reviewer #1 (Remarks to the Author):

This manuscript reports lower apparent temperature sensitivity of heterotrophic respiration (Rh) in warmed compared to control grassland plots. Functional gene abundance was measured and used in a model that incorporated microbial enzymatic processes. They attribute the observed change of temperature sensitivity to an inferred thermal acclimation of the soil microbial community. They then extrapolate their findings to all grasslands of the globe to suggest that a similar warming would reduce soil carbon losses by about 0.5 Pg C/yr.

While the effort to include gene expression in a soil carbon decomposition model is laudable, I have several major qualms about this manuscript.

1. When the authors introduce this topic to the reader, they make the unfortunate common mistake of referring to all instances of a “dampened effect of warming” as “respiratory acclimation” (lines 78-80). They go on to discuss mechanisms of acclimation (lines 85-87), including several distinct processes under that umbrella term, which actually are not forms of acclimation at all.

“Acclimation” is a term derived from organismal physiology, describing how an organism can alter (acclimate) its metabolic processes in response to a changing environment. However, the soil microbial community is not a single organism. It is possible that individual bacteria and other soil organisms could acclimate to changing temperature by producing isoenzymes, for example, that perform the same function but with different properties, such as temperature sensitivity. However, it is equally or more likely that the composition of the soil microbial community may shift as temperature change, in which case the organisms are not acclimating, but rather some taxa die out and are replaced by others. This is not acclimation. Likewise, exhaustion of readily available substrates may result in decreased Rh and an apparent decline in temperature sensitivity because lack of substrate prevents the expression of the enzyme’s intrinsic temperature sensitivity, but this is not acclimation either. Rather, it is simply a substrate limitation that prevents maximum enzymatic activity and that masks the enzyme’s intrinsic temperature sensitivity. Unfortunately, the text in this section will confuse readers on a topic that is already complex. A better umbrella term that includes organismal acclimation and change in community structure is “thermal adaptation” (see Bradford et al. 2019. *Nature Ecology & Evolution*, 3: 223–231), but substrate supply is still in its own category.

2. The authors note in lines 131-134 that the treatment changed the plant community structure. This is likely the cause of changing soil microbial structure and gene expression, because the quantity and quality of C input from the plants differ depending upon the species. Hence, the change in microbial community structure may not have been a direct effect of a response to the temperature treatment, but rather due to the direct effect of changing plant community structure, which was, in turn, affected by the warming treatment.

3. On lines 149-151, the authors note a decline (albeit non-significant) in autotrophic respiration (Ra) in the warming treatment, and later speculate that “soil C input in the form of plant litter may substantially contribute to the stability of soil C when plant roots were excluded.” However, most grassland ecologist would argue that grassland soils are rhizosphere soils, dominated by root inputs. Were the tubes left in the same place in the soil for all 7 years of the experiment?

Eliminating root inputs for that long of a period surely would have affects both Ra and Rh, as the root inputs are necessary for fueling Rh. A soil without root inputs for several years will not have a natural Rh. If they inserted the tube anew each year, then there is another artifact caused by severing roots and increasing heterotrophic respiration using the dead roots as substrates. I suggest that the authors review Savage et al. (2019. *Biogeochemistry* <https://doi.org/10.1007/s10533-018-0472-8>) for analysis of artifacts of the trenching method for separating Rh and Ra, and include some discussion of these sources of error in their analysis.

4. The term “intrinsic” is first used in line 173, but it is not defined. A rather confusing definition of sorts is offered in the methods section (lines 420-423): “unconfounded temperature sensitivity.” I’m not sure if that is the same as the common usage, sensu Davidson & Janssens (2006), that intrinsic temperature sensitivity is determined by the inherent activation energy of an enzyme that is not limited by substrate supply or other limitations. I also don’t understand the term

“constrained Q10” on lines 421-422. My guess is that the MEND model was constrained by observed (apparent) Q10 values and the activation energies or the inherent Q10s of the enzyme activities within the model were then fit to the data. Assuming that the model structure includes all other factor that could be temperature sensitive, such as diffusion of substrate and confounding effects of temperature and moisture content, then this approach could provide a reasonable estimate of the intrinsic temperature sensitivities of the enzymes. However, it is not clear what factors within the model reduced these fitted “intrinsic” temperature sensitivities of the enzymes to values below the observed apparent temperature sensitivities. I see from Fig. 1b that soil moisture was significantly lower in the warmed plots, so substrate diffusion through thinner soil water films may have limited Rh, but I don’t know how MEND handles substrate diffusion. Too little information is provided about how the modeling was done and how factors that are confounded with temperature, and hence affect the difference between intrinsic and apparent temperature sensitivities, were parsed out. Actually, I would expect that lower soil moisture would cause the apparent temperature sensitivity to be lower than the intrinsic temperature sensitivity, so the opposite result is very puzzling. It could be that we are defining “intrinsic” differently or that the MEND model is simulating important processes that are not adequately explained. For example, line 275 claims that the MEND model “can remove confounding effects of other environmental factors” but there is no explanation that I can find as to what those factors were and how important they were. Even if I were to read about the MEND model in another paper, I still wouldn’t know how it handled confounding environmental variables and produced the lower estimated intrinsic Q10 values in the present study.

5. I am also puzzled how the effects of changing microbial community and gene expression can be assessed with only annual soil sampling. We know that microbial communities change seasonally and that many important processes occur some seasons and other processes in other seasons, whereas these results are based entirely on samples collected once per year in the autumn. Why is maximum plant biomass necessarily the most important time for soil microbial processes? For example, perhaps there is a more important time for decomposition when soils are warming but still wet in the late spring. Indeed, Fig. S2 shows peak respiration in the spring in some years and in mid-summer in most years, not in the early autumn when plant biomass is at its peak.

6. Available C substrates are a tiny fraction of total soil C stocks, so, contrary to the reasoning presented in lines 193-196, one would not expect a detectable change in total soil C if available C substrates became limiting. The increase in NEE in the warming plots could be due to accumulation of C as undecomposed leaf litter or live or dead root biomass and may have nothing to do with increasing available substrate for soil heterotrophic respiration or avoiding Rh substrate limitation. I don’t think that substrate limitation can be ruled out. Moreover, I’m willing to bet that the calculated difference in Rh inferred to be due to “acclimation” would be an undetectable change in soil C stocks given the size and spatial heterogeneity of soil C stocks.

7. It is not surprising that as the number of model parameters increases from the TECO to the tMEND to the gMEND models, that the model performance also improves. One can often get a better fit to the data by making a model more complex with more parameters to fit, but that doesn’t mean that we have greater confidence that the model gets the right answer for the right reason. To understand the usefulness of adding model complexity, the models should be compared using the Akaike information criterion (AIC) or some similar approach. Without that, the comparisons among models in lines 252-283 are not very meaningful.

8. I cannot agree with the statement in lines 272-273 that “Q10 values below 2 are preferred for better global C cycle modeling.” That may be true for apparent Q10 values in many situations, because there are confounding factors, such as substrate limitation or water stress, that prevent higher intrinsic Q10 values from being expressed. However, the most important enzymes tend to have activation energies that translate to intrinsic Q10s near 2 or slightly above in temperature ranges of about 5 – 15 C.

9. Extrapolation of these results to global grasslands is too much of a stretch for me. Grasslands are very diverse, spanning from the tropics to the temperate zone, including xeric and mesic climates, occurring on a diversity of soil types, with variation in C3 versus C4 dominance, and with differing effects of grazing, etc. This work was done in one site on a sandy soil in a temperate region (without grazing, I presume), which is not necessarily representative of the broad spectrum

of grassland conditions. Hence, the extrapolation of a change in soil C loss of 0.49 PgC/yr (line 293) is an over-the-top extrapolation, in my opinion, and should be removed.

10. In Fig. S4, I suggest adding horizontal and vertical error bars for each point, so that we can see how variable these central values are relative to their position near the 1:1 line.

Reviewer #2 (Remarks to the Author):

This manuscript integrates measurements and modeling of a grassland warming experiment to investigate how temperature sensitivity of heterotrophic soil respiration changed under warming. The integration of multiple measurement techniques, including metagenomic and substrate degradation measurements, with microbial-explicit modeling is powerful and novel. The use of GeoChip data in model calibration was a smart strategy for integrating microbial community function data with a model. Integrating this type of data with microbial-explicit models has been a challenge for the field and this is a promising solution. This combination provides a very intriguing convergence of evidence supporting the main result that the microbial community shifted over the experiment in a way that drove acclimation of temperature sensitivity. The changes in Q10 do seem robust, appearing in both observational analysis and the calibrated model.

I reviewed a previous version of this manuscript (in **[REDACTED]**) and the current manuscript has addressed most of the issues that I found in my previous review. It is well written, well organized, and easy to follow and the conclusions are generally well supported by measurements and statistical analyses.

One area where I think there is still room for improvement is the role of soil moisture. There was a significant decrease (6%) in soil moisture in the warming treatment, which potentially contributed to a decline in respiration and its temperature sensitivity. In the previous manuscript that I reviewed, the study used what I felt was an inappropriate statistical analysis to argue that moisture was not the main driver. In this version, the manuscript removed that analysis but now does not include any discussion of the potential role of soil moisture. There is clear evidence from previous studies that soil moisture affects soil respiration (e.g., Davidson and Janssens 2006, Moyano et al., 2013, Manzoni et al 2012), and I think the potential for this effect cannot be ignored. In fact, the structural equation model (Fig. 2b) suggested that moisture was a more direct driver than temperature of microbial community shifts. It may not be possible to quantify the impact of soil moisture on respiration in this experiment from the measurements alone. If so, the observational limitation simply be acknowledged in the paper and soil moisture could be discussed as a potentially important factor. The MEND model has been previously applied in moisture-related contexts (e.g., Wang et al., 2015). Perhaps the model could be used to estimate the potential role of soil moisture, by conducting warming simulations with or without drying decreased soil moisture. This would provide a basis for discussing whether drying was a key driver of the respiration response. But overall, I don't think the moisture change can just be ignored when interpreting the results of this experiment.

Second, an issue I also brought up in my previous review is that model output and respiration measurements do not seem to be available anywhere. This would be particularly helpful for evaluating and interpreting the results related to changing Q10 values. It would be helpful to have figures in the supplement showing the respiration vs temperature relationship and Q10 model fit line for each year in warmed and control treatments. Some of the Q10 fits are statistically significant but have low R2 (Table S1), and being able to visualize those fits is important context for interpreting the results of the paper. Ideally, model output and respiration measurements would be available as raw data or spreadsheets (machine-readable formats) in the interest of open data and reproducibility. The current manuscript does not provide information on how many total respiration measurements there were or when they were taken beyond vague information ("measured once or twice a month"). For figures like Figure 3b, this makes it impossible to tell

how many observations are being shown or how they were distributed in time. A table that included every respiration measurement and the date it was collected, along with soil temperature and moisture associated with the measurement, in a format that could be downloaded and analyzed, would solve this problem and support discoverability and reproducibility according to current scientific standards. "Available from the author upon request" is not a good solution to this issue.

Specific comments:

Line 62: It is ambiguous what "whose" refers to in this sentence. This sentence should be reworded.

Line 110-111: "right after the continuous warming" needs rewording

Line 272: "preferred for better global C cycle modeling" could be edited to be more specific about why these numbers are preferred, i.e. they match better with observation-based estimates

Line 275-276: The statement that using MEND can remove confounding effects of other environmental factors would be a good place to introduce the idea of using the model to estimate the contribution of changing soil moisture vs changing temperature to changes in respiration

Line 460: It's not clear how weighting factors were assigned

Line 753-757: The explanation of how the bars were calculated is difficult to understand and is pretty complex to be in a figure caption. Consider rewriting or simplifying.

References cited:

Davidson, E. A., and Janssens, I. A. (2006). Temperature sensitivity of soil carbon decomposition and feedbacks to climate change. *Nature* 440, 165–173. Available at: [papers2://publication/doi/10.1038/nature04514](https://doi.org/10.1038/nature04514).

Manzoni, S., Schimel, J. P., and Porporato, A. (2012). Responses of soil microbial communities to water stress: results from a meta-analysis. *Ecology* 93, 930–938. doi:10.1890/11-0026.1.

Moyano, F. E., Manzoni, S., and Chenu, C. (2013). Responses of soil heterotrophic respiration to moisture availability: An exploration of processes and models. *Soil Biol. Biochem.* 59, 72–85. doi:10.1016/j.soilbio.2013.01.002.

Wang, G., Jagadamma, S., Mayes, M. et al. Microbial dormancy improves development and experimental validation of ecosystem model. *ISME J* 9, 226–237 (2015).

<https://doi.org/10.1038/ismej.2014.120>

Responses to Comments from Reviewers

We have addressed the reviewers' comments and questions point-by-point. The original reviewer's comments are italicized and our responses to the reviewer's comments follow. The numbers of lines in the text are referred to the revised version, where corrections are tracked.

A. Responses to Reviewer #1 (Remarks to the Author):

A1. Thermal adaptation vs. acclimation

When the authors introduce this topic to the reader, they make the unfortunate common mistake of referring to all instances of a “dampened effect of warming” as “respiratory acclimation” (lines 78-80). They go on to discuss mechanisms of acclimation (lines 85-87), including several distinct processes under that umbrella term, which actually are not forms of acclimation at all. “Acclimation” is a term derived from organismal physiology, describing how an organism can alter (acclimate) its metabolic processes in response to a changing environment. However, the soil microbial community is not a single organism. It is possible that individual bacteria and other soil organisms could acclimate to changing temperature by producing isoenzymes, for example, that perform the same function but with different properties, such as temperature sensitivity. However, it is equally or more likely that the composition of the soil microbial community may shift as temperature change, in which case the organisms are not acclimating, but rather some taxa die out and are replaced by others. This is not acclimation. Likewise, exhaustion of readily available substrates may result in decreased Rh and an apparent decline in temperature sensitivity because lack of substrate prevents the expression of the enzyme's intrinsic temperature sensitivity, but this is not acclimation either. Rather, it is simply a substrate limitation that prevents maximum enzymatic activity and that masks the enzyme's intrinsic temperature sensitivity. Unfortunately, the text in this section will confuse readers on a topic that is already complex. A better umbrella term that includes organismal acclimation and change in community structure is “thermal adaptation” (see Bradford et al. 2019. Nature Ecology & Evolution, 3: 223–231), but substrate supply is still in its own category.

Responses: Thank you for the excellent suggestions. We agree that “thermal adaptation” is a better umbrella term than “acclimation” in this study, since the thermal adaptation of soil respiration may occur through individual acclimatization and shifts of soil microbial communities. The corresponding changes had been made throughout the manuscript, and more references about thermal adaptation had been added in the revised manuscript.

A2. Effect of changing plant community

The authors note in lines 131-134 that the treatment changed the plant community structure. This is likely the cause of changing soil microbial structure and gene expression, because the

quantity and quality of C input from the plants differ depending upon the species. Hence, the change in microbial community structure may not have been a direct effect of a response to the temperature treatment, but rather due to the direct effect of changing plant community structure, which was, in turn, affected by the warming treatment.

Responses: Thank you for the constructive suggestion. We agree that warming-induced changes in the plant community structure could significantly contribute to the shifts of soil microbial composition and functional structure, since the quantity and quality of soil C input from the plant differ depending upon the species. Structural equation modeling (SEM)-based analysis showed that C₃ plant biomass was a direct driver to change soil microbial functional structure under the long-term warming. Therefore, to address the reviewer's comments, more results and discussion about the effects of changing plant community had been added in the revised manuscript as: "In addition, warming can also alter soil microbial community structure indirectly through changing plant community structure, because the quantity and quality of soil C input from the plants differ depending upon the species. In this study, C₃ plant biomass was significantly ($p < 0.05$) decreased by warming and exhibited a direct effect on soil microbial function structure in the SEM-based analysis (Fig. 2b). All of these results indicated that the adaptive changes in microbial community composition and functional structure resulted from the combined effects of the increase of soil temperature, decrease of soil moisture, and changing plant community structure under long-term warming" (Line 254-262).

A3. Artifacts of methods to partition R_h from R_t

On lines 149-151, the authors note a decline (albeit non-significant) in autotrophic respiration (R_a) in the warming treatment, and later speculate that "soil C input in the form of plant litter may substantially contribute to the stability of soil C when plant roots were excluded." However, most grassland ecologist would argue that grassland soils are rhizosphere soils, dominated by root inputs. Were the tubes left in the same place in the soil for all 7 years of the experiment? Eliminating root inputs for that long of a period surely would have affects both R_a and R_h , as the root inputs are necessary for fueling R_h . A soil without root inputs for several years will not have a natural R_h . If they inserted the tube anew each year, then there is another artifact caused by severing roots and increasing heterotrophic respiration using the dead roots as substrates. I suggest that the authors review Savage et al. (2019. Biogeochemistry <https://doi.org/10.1007/s10533-018-0472-8>) for analysis of artifacts of the trenching method for separating R_h and R_a , and include some discussion of these sources of error in their analysis.

Responses: We understand the reviewer's concerns. In this study, the deep (70 cm) PVC tubes for measuring soil R_h were permanently left in the same place of the soil. We agree that the root exclusion method by deep PVC tubes for partitioning R_a and R_h had some potential artifacts, including soil moisture and temperature changes, exclusions of plant detritus inputs, and soil

microbial community changes. And we have carefully reviewed the analysis of artifacts caused by the trenching method for separating R_h and R_a . Unfortunately, we can not further quantify the artifacts of root exclusion method in our study, since soil moisture, temperature and severing roots in the deep PVC tubes could not be measured in the 7 years of warming. Although the artifacts exist for the root exclusion method, such artifacts would be less problematic by focusing on the relative changes of soil respiration between treatments and control, which is the major focus of this study. To address the reviewer's comments, we had added more discussion about the sources of errors in our measurements in the revised manuscript as "Notably, the root exclusion method by deep PVC tubes for partitioning R_a and R_h had some potential artifacts, including soil moisture and temperature changes, exclusions of plant detritus inputs, and soil microbial community changes, although these artifacts may be less problematic when we focused on the relative changes between treatments and controls. Soil moisture and temperature in the deep PVC tubes were not measured in this study, but previous study indicated that the similar root exclusion method (trenching method) can artificially increase soil moisture and temperature, and thus could overestimate soil R_h . Soil microbial community structure and biomass may not be significantly changed by root exclusion, as revealed by a previous study. The severing roots by inserting PVC tube in soil may result in a transient increase of soil respiration. In this study, soil R_h was firstly measured at least 8 months after the insertion of PVC tube into soil, so the effects of decomposition of severing roots on measured R_h should be minimized. However, it is highly possible that the exclusion of root inputs to soil as dead roots and root exudates for a long time could underestimate soil R_h , and in turn overestimate soil R_a " (Line 158-171)

A4. Modeling analysis

The term "intrinsic" is first used in line 173, but it is not defined. A rather confusing definition of sorts is offered in the methods section (lines 420-423): "unconfounded temperature sensitivity." I'm not sure if that is the same as the common usage, sensu Davidson & Janssens (2006), that intrinsic temperature sensitivity is determined by the inherent activation energy of an enzyme that is not limited by substrate supply or other limitations. I also don't understand the term "constrained Q_{10} " on lines 421-422. My guess is that the MEND model was constrained by observed (apparent) Q_{10} values and the activation energies or the inherent Q_{10} s of the enzyme activities within the model were then fit to the data. Assuming that the model structure includes all other factor that could be temperature sensitive, such as diffusion of substrate and confounding effects of temperature and moisture content, then this approach could provide a reasonable estimate of the intrinsic temperature sensitivities of the enzymes. However, it is not clear what factors within the model reduced these fitted "intrinsic" temperature sensitivities of the enzymes to values below the observed apparent temperature sensitivities. I see from Fig. 1b that soil moisture was significantly lower in the warmed plots, so substrate diffusion through thinner soil water films may have limited R_h , but I don't know how MEND handles substrate diffusion. Too little information is provided about how the modeling was done and how factors

that are confounded with temperature, and hence affect the difference between intrinsic and apparent temperature sensitivities, were parsed out. Actually, I would expect that lower soil moisture would cause the apparent temperature sensitivity to be lower than the intrinsic temperature sensitivity, so the opposite result is very puzzling. It could be that we are defining “intrinsic” differently or that the MEND model is simulating important processes that are not adequately explained. For example, line 275 claims that the MEND model “can remove confounding effects of other environmental factors” but there is no explanation that I can find as to what those factors were and how important they were. Even if I were to read about the MEND model in another paper, I still wouldn’t know how it handled confounding environmental variables and produced the lower estimated intrinsic Q_{10} values in the present study.

Responses: Thanks for pointing these issues out. We use the term “intrinsic Q_{10} ”, as used in Mahecha et al. (2010) and Zhou et al. (2017), to represent “direct response” to temperature versus the confounded effects of multiple factors, such as soil moisture and substrate availability. Similar to other soil biogeochemical models, the process-based MEND model uses different response functions (see Table S5) to represent the effects of soil pH, temperature, and moisture on various transformation processes (e.g., soil organic matter (SOM) decomposition, dissolved organic matter (DOM) sorption-desorption, microbial growth, maintenance, mortality, and dormancy). Therefore, in the model structure of MEND, the direct effect of soil temperature is distinguished from the effects of other environmental factors, which serves as the purpose to “remove” the confounding effects of multiple factors. To address the reviewer’s comments, we added more description about the intrinsic Q_{10} in the main text as “Here we used the “intrinsic temperature sensitivity” to distinguish the MEND-derived Q_{10} from the “apparent Q_{10} ” estimated from the relationship between respiration and temperature. The intrinsic temperature sensitivity represents “direct response” of heterotrophic respiration to temperature change in the modeling context, as we used different response functions in MEND to represent the direct effects of soil pH, temperature, and moisture on various transformation processes” (Line 188-193).. Furthermore, we added a Table S5 to provide more detailed information about model structure of MEND to handle confounding environmental variables, such soil pH, moisture and temperature.

The term “Constrained Q_{10} ” means the intrinsic Q_{10} is derived by model fitting constrained by available observations including respiration and gene abundances. Instead of directly simulating the substrate diffusion process, the MEND model describes the effects of soil moisture on microbial and enzyme activities that subsequently affect SOM decomposition and DOM availability for microbial growth and maintenance.

In addition to temperature and moisture, other factors (e.g., substrate availability and physicochemical protection) will also affect the apparent temperature sensitivity (Davidson & Janssens, 2006). Therefore, it’s uncertain to conclude whether intrinsic Q_{10} is higher or lower than apparent Q_{10} without a quantitative approach to separate the confounding effects. In this

study, we derive intrinsic Q_{10} from inverse process-based modeling and apparent Q_{10} by the observed relationship between respiration and soil temperature. There could be uncertainties in these estimates, however, we show that the intrinsic Q_{10} is lower than the apparent Q_{10} , consistent with previous site-level study by Zhou et al. (2017) and global-scale study by Mahecha et al. (2010). To address the reviewer's comments, we rewritten the results of MEND analysis as "As the MEND model uses different response functions to represent the effects of soil pH, temperature, and moisture on various transformation processes (Table S5), the Q_{10} in the MEND model solely reflects the microbial and enzymatic responses to temperature change. This means that the direct effect of soil temperature can be distinguished from the effects of other environmental factors. To discern the effects of soil temperature from moisture, we used gMEND to estimate the R_h response to a single-factor change in soil temperature or moisture during the seven year's experimental period. Compared to the MEND-simulated mean R_h under control, changing soil temperature under warming would result in a 22.2% increase in R_h , whereas changing soil moisture would cause a decrease in R_h by 8.1% (Fig. S14). Although the negative effect on R_h due to slightly drier soil under warming was considerable, it was completely shifted by the significant positive effect from soil temperature increase" (Line 315-325).

References:

- Davidson, E.A., Janssens, I.A., 2006. Temperature sensitivity of soil carbon decomposition and feedbacks to climate change. *Nature*, 440(7081): 165.
- Mahecha, M.D. et al., 2010. Global convergence in the temperature sensitivity of respiration at ecosystem level. *Science*, 329(5993): 838-840.
- Zhou, X., Xu, X., Zhou, G., Luo, Y., 2018. Temperature sensitivity of soil organic carbon decomposition increased with mean carbon residence time: Field incubation and data assimilation. *Global change biology*, 24(2): 810-822.

A5. Assessment timing of microbial community

I am also puzzled how the effects of changing microbial community and gene expression can be assessed with only annual soil sampling. We know that microbial communities change seasonally and that many important processes occur some seasons and other processes in other seasons, whereas these results are based entirely on samples collected once per year in the autumn. Why is maximum plant biomass necessarily the most important time for soil microbial processes? For example, perhaps there is a more important time for decomposition when soils are warming but still wet in the late spring. Indeed, Fig. S2 shows peak respiration in the spring in some years and in mid-summer in most years, not in the early autumn when plant biomass is at its peak.

Response: We agree that microbial communities change along seasons and that different microbial processes may occur in different seasons. The effects of changing microbial community and gene expression provides only a snapshot of soil microbial community in the

sampling time. However, high time-resolution studies about soil microbial communities are not feasible for most of manipulated, field experiment sites. The major scientific limitation of these types of studies is the destructive nature of sampling, and hence the majority of long-term experiment sites do not allow periodic soil sampling due to small plot size. Picturing a small plot being probed time after time to excise soil samples, the plot will eventually become a “swiss cheese” with too many holes to represent a natural soil ecosystem. Actually, our experiment site is one of the very few that perform annual soil sampling, since we carefully design a sampling strategy and map every sample event. Also, this study does not look for microbial gene expression, which has very short half-life time (3-5 mins for mRNAs). Instead, we focused on the mean population changes, which relied on DNA-based molecular analyses. Although the information derived from annual soil samples in the autumn can't reflect the responses of soil microbial communities to warming in other seasons, comparing the **mean** effects across different treatments should be still meaningful as demonstrated by this study and several of previous studies (Johnston et al. 2019; Xue et al. 2016; Zhou et al. 2012). Also, many environmental variables (i.e., soil temperature, moisture), soil respirations and ecosystem C fluxes (i.e., GPP, NEE, ER) were monitored daily or monthly, which were useful to assess soil microbial contribution to C cycling in different seasons based on modeling analysis.

In order to compare the responses of soil microbial communities to warming, annual soil samples should be collected at the similar time every year. In this study, we chose to collect soil samples to analyze soil microbial processes at the maximum plant biomass in the autumn, since the activities of both soil microbial communities and plant communities are greatest at this time, and the annual **mean** effects should be more representative by the samples at this time point. We agree that there are other important periods for soil C decomposition, soil C input or other C cycling processes. Unfortunately, we can only collect soil samples once every year to analyze warming effects on soil microbial communities in this long-term warming experiment site. Hopefully, higher time-resolution studies can be performed in future to provide comprehensive insights on the responses of soil microbial communities to climate warming over time.

References:

- E. R. Johnston *et al.* 2019. Responses of tundra soil microbial communities to half a decade of experimental warming at two critical depths. *Proceedings of the National Academy of Sciences*, 116(30): 15096-15105.
- Zhou, J., Xue, K., Xie, J. et al. 2012. Microbial mediation of carbon-cycle feedbacks to climate warming. *Nature Clim Change* 2: 106–110.
- Xue, K., M. Yuan, M., J. Shi, Z. et al. 2016. Tundra soil carbon is vulnerable to rapid microbial decomposition under climate warming. *Nature Clim Change* 6: 595–600.

A6.

Available C substrates are a tiny fraction of total soil C stocks, so, contrary to the reasoning presented in lines 193-196, one would not expect a detectable change in total soil C if available C substrates became limiting. The increase in NEE in the warming plots could be due to accumulation of C as undecomposed leaf litter or live or dead root biomass and may have nothing to do with increasing available substrate for soil heterotrophic respiration or avoiding Rh substrate limitation. I don't think that substrate limitation can be ruled out. Moreover, I'm willing to bet that the calculated difference in Rh inferred to be due to "acclimation" would be an undetectable change in soil C stocks given the size and spatial heterogeneity of soil C stocks.

Response: We fully agree that available C substrates are a tiny fraction of total soil C stocks, and have a rapid turnover. However, our BILOG results revealed that, after 7 years of warming, microbial metabolism underpinning the utilization ability of most available C substrates were considerably higher under warming than control. If warming induced soil C substrate depletion, soil microbial community should shift to grow on recalcitrant C substrates that are less energetically efficient (Bradford et al. 2019. *Nature Ecology & Evolution*, 3: 223–231). But why did soil microbial community have the higher utilization ability of available C substrates after long-term warming? One reasonable explanation is that soil C stocks, especially labile C pools as the sources of available C substrates were relatively stable and provided equal or more available C substrates for soil microbial community after long-term warming, compared with controls. NEE data can support this note, since higher NEE suggested that soil labile and/or recalcitrant C input as plant litter, root biomass or exudates should be similar or even higher under warming than control. Some of the plant litter, root biomass or exudates must have been decomposed into soil available substrates for soil heterotrophic respiration in the 7-years of experiment, especially in a grassland ecosystem with a rapid C turnover. In addition, we agree that the difference in soil R_h due to thermal adaptation may be an undetectable change in soil C stocks given the size and spatial heterogeneity of soil C stocks. However, warming-induced changes of soil C stocks were significant in some studies (Crowther et al. 2016. *Nature*, 540: 104–108). Therefore, the unchanged soil total C in this study suggested the stability of soil total carbon under the long-term warming, which did not support the substrate depletion. We agree that substrate limitation can not be completely ruled out in this study, but warming-induced substrate depletion is not consistent with most of our evidences in this study. Therefore, to address Reviewer's concern, we made the corresponding changes in the revised manuscript as "In this study, several lines of evidence suggest that the decreased temperature sensitivity of microbial respiration was not mainly due to warming-induced substrate depletion. First, available C substrates are a tiny fraction of total soil C stocks and have rapid turnovers, but our BILOG results revealed that, after 7 years of warming, microbial metabolism underpinning the utilization ability of most available C substrates were considerably higher under warming than control (Fig. S5). A reasonable explanation for the result is that soil C stocks, especially labile C pools as the

sources of available C substrates were relatively stable without substantial reduction, and can provide equal or more available C substrates after long-term warming, compared to the controls. Second, NEE was higher under warming than control (Fig. 1c), suggesting that more soil labile and/or recalcitrant C input as plant litter, root biomass or exudates counteracted the consumption of soil available C substrates by microbial respiration. Third, the unchanged annual soil C from 2010 to 2016 (Fig. S1c) indicated the stability of soil total C under the long-term warming, which does not support the expectation garnered from the substrate depletion hypothesis. Altogether, these results suggested that the turnover of soil labile C may be accelerated by warming, but warming did not lead to the depletion of soil labile C. Therefore, the reduced temperature sensitivity of soil respiration appears to be less likely due to warming-induced substrate depletion, although the effects of substrate depletion could not be completely ruled out” (Line 204-221).

A7.

It is not surprising that as the number of model parameters increases from the TECO to the tMEND to the gMEND models, that the model performance also improves. One can often get a better fit to the data by making a model more complex with more parameters to fit, but that doesn't mean that we have greater confidence that the model gets the right answer for the right reason. To understand the usefulness of adding model complexity, the models should be compared using the Akaike information criterion (AIC) or some similar approach. Without that, the comparisons among models in lines 252-283 are not very meaningful.

Responses: We calibrated 10 TECO parameters and 11 MEND parameters. The AIC results show that the MEND performance is better than TECO in both cases (control and warmed). To address the reviewer's comments, we added the AIC results in the revised manuscript as “We calibrated 10 parameters for TECO and 11 parameters for tMEND and gMEND, the Akaike information criterions (AIC) of the MEND models (-14.55 for warming and -38.30 for control) were smaller than those of the TECO model (-4.14 for warming and -34.79 for control), suggesting a better fit by the MEND model” (Line 334-338).

References

Liang, J. et al., 2019. Evaluating the E3SM Land Model version 0 (ELMv0) at a temperate forest site using flux and soil water measurements. *Geoscientific Model Development* 12: 1601-1612. DOI:10.5194/gmd-12-1601-2019

A8.

I cannot agree with the statement in lines 272-273 that “Q10 values below 2 are preferred for better global C cycle modeling.” That may be true for apparent Q10 values in many situations, because there are confounding factors, such as substrate limitation or water stress, that prevent

higher intrinsic Q_{10} values from being expressed. However, the most important enzymes tend to have activation energies that translate to intrinsic Q_{10} s near 2 or slightly above in temperature ranges of about 5 – 15 C.

Responses: We understand the reviewer’s concern. Here we mean that Q_{10} values near 2 are often used in most cases. Therefore, we rewritten this sentence as “In addition, the MEND-derived intrinsic Q_{10} values were confined from 1.20–2.42 (tMEND) to a narrow range of 1.27–2.13 (gMEND), corroborating that Q_{10} values of 2 or below are usually used in global C cycle modeling. The intrinsic Q_{10} values also concurred with previous site-level and global-scale studies” (Line 338-341).

Based on commonly used definition of “intrinsic Q_{10} ” in the context of process-based ecosystem models, the “intrinsic Q_{10} ” used in most models are usually ≤ 2 (Mahecha, et al., 2010; Mayer et al., 2018), for example, $Q_{10} = 1.5$ for SOM decomposition in CLM (Lawrence et al., 2019). Our previous study showed that average $Q_{10} = 1.8, 1.6, 1.6, 2.1,$ and 2.1 for five main ligninases and cellulases with a temperature increase from 20 to 30°C (Wang et al. 2012). Noting that these Q_{10} values were mostly derived from purified enzymes. The “intrinsic Q_{10} ” (i.e., temperature-only sensitivity) in field conditions could be different from the intrinsic Q_{10} of purified enzymes.

References

- Lawrence, D. et al., 2019. Technical description of version 5.0 of the Community Land Model (CLM). National Center for Atmospheric Research (NCAR), Boulder CO, pp. 329.
- Mahecha, M.D. et al., 2010. Global convergence in the temperature sensitivity of respiration at ecosystem level. *Science*, 329(5993): 838-840.
- Meyer, N., Welp, G., Amelung, W., 2018. The temperature sensitivity (Q_{10}) of soil respiration: controlling factors and spatial prediction at regional scale based on environmental soil classes. *Global Biogeochemical Cycles*, 32(2): 306-323.
- Wang, G., Post, W.M., Mayes, M.A., Frerichs, J.T., Jagadamma, S., 2012. Parameter estimation for models of ligninolytic and cellulolytic enzyme kinetics. *Soil Biology and Biochemistry*, 48: 28-38. DOI:10.1016/j.soilbio.2012.01.011

A9.

Extrapolation of these results to global grasslands is too much of a stretch for me. Grasslands are very diverse, spanning from the tropics to the temperate zone, including xeric and mesic climates, occurring on a diversity of soil types, with variation in C3 versus C4 dominance, and with differing effects of grazing, etc. This work was done in one site on a sandy soil in a temperate region (without grazing, I presume), which is not necessarily representative of the broad spectrum of grassland conditions. Hence, the extrapolation of a change in soil C loss of 0.49 PgC/yr (line 293) is an over-the-top extrapolation, in my opinion, and should be removed.

Response: Thanks for pointing the issues out. We agree that extrapolation of these results to global grasslands may lead to large estimation errors, so we removed this extrapolation in this study.

A10.

In Fig. S4, I suggest adding horizontal and vertical error bars for each point, so that we can see how variable these central values are relative to their position near the 1:1 line. .

Response: Thanks for your insightful suggestion. We have added horizontal and vertical error bars for each point in the Fig. S4 of the revised manuscript (Fig. S5).

B. Responses to Reviewer #2 (Remarks to the Author):

B1.

This manuscript integrates measurements and modeling of a grassland warming experiment to investigate how temperature sensitivity of heterotrophic soil respiration changed under warming. The integration of multiple measurement techniques, including metagenomic and substrate degradation measurements, with microbial-explicit modeling is powerful and novel. The use of GeoChip data in model calibration was a smart strategy for integrating microbial community function data with a model. Integrating this type of data with microbial-explicit models has been a challenge for the field and this is a promising solution. This combination provides a very intriguing convergence of evidence supporting the main result that the microbial community shifted over the experiment in a way that drove acclimation of temperature sensitivity. The changes in Q10 do seem robust, appearing in both observational analysis and the calibrated model.

I reviewed a previous version of this manuscript (in [REDACTED]) and the current manuscript has addressed most of the issues that I found in my previous review. It is well written, well organized, and easy to follow and the conclusions are generally well supported by measurements and statistical analyses.

Response: Thank you very much for the complimentary comments.

B2. Soil moisture

One area where I think there is still room for improvement is the role of soil moisture. There was a significant decrease (6%) in soil moisture in the warming treatment, which potentially contributed to a decline in respiration and its temperature sensitivity. In the previous manuscript that I reviewed, the study used what I felt was an inappropriate statistical analysis to argue that moisture was not the main driver. In this version, the manuscript removed that analysis but now

does not include any discussion of the potential role of soil moisture. There is clear evidence from previous studies that soil moisture affects soil respiration (e.g., Davidson and Janssens 2006, Moyano et al., 2013, Manzoni et al 2012), and I think the potential for this effect cannot be ignored. In fact, the structural equation model (Fig. 2b) suggested that moisture was a more direct driver than temperature of microbial community shifts. It may not be possible to quantify the impact of soil moisture on respiration in this experiment from the measurements alone. If so, the observational limitation simply be acknowledged in the paper and soil moisture could be discussed as a potentially important factor. The MEND model has been previously applied in moisture-related contexts (e.g., Wang et al., 2015). Perhaps the model could be used to estimate the potential role of soil moisture, by conducting warming simulations with or without drying decreased soil moisture. This would provide a basis for discussing whether drying was a key driver of the respiration response. But overall, I don't think the moisture change can just be ignored when interpreting the results of this experiment.

Responses: Thanks for the insightful suggestions. We agree that more discussion of the potential role of soil moisture can provide more comprehensive insights into the responses of soil respiration to warming and the underlying microbial mechanisms. We now provide more detailed results and discussion on the role of soil moisture in affecting soil microbial function structure and R_h . Our results indicated that warming significantly decreased soil moisture by 6%. Furthermore, strong and significant correlations were observed between soil moisture and microbial composition and functional structure as revealed by the Mantel test (Fig. 2a and S6) and canonical correspondence analyses (CCA) (Fig. S7). Structural equation modeling (SEM)-based analysis indicated that soil moisture significantly affected soil R_h through shifting microbial functional structure. We also agree that soil moisture may play more important roles than temperature in changing soil microbial function structure and soil respiration in some case. In the extremely drought year 2011, it is highly possible that severe soil moisture limitation led to no significant temperature sensitivities of soil microbial respiration observed in the year. The corresponding changes had been made in the manuscript as: “However, the shifts of microbial communities and soil R_h may not be solely explained by the rising temperature under warming, since significant decreases of soil moisture were observed under warming, and strong correlations occurred between soil moisture and microbial composition and functional structure (Fig. 1b and 2a). Previous studies provided clear evidences that soil moisture limitation can weaken the stimulation of warming on soil respiration. Congruously, our SEM-based analysis suggested that soil moisture significantly ($p < 0.05$) affected soil R_h through shifting microbial functional structure. It is highly possible that severe soil moisture limitation played more important role in changing soil microbial community and R_h than temperature in the extremely drought year (2011), which led to no significant temperature sensitivities of soil microbial respiration observed in the year (Fig. S3)” (Line 245-254).

We also used MEND to quantify the impact of soil temperature or moisture on R_h alone. The results indicated that the effects of soil temperature were much larger than the effects of soil

moisture. The corresponding changes had been made in the manuscript as: *“To discern the effects of soil temperature from moisture, we used gMEND to estimate the R_h response to a single-factor change in soil temperature or moisture during the seven year’s experimental period. Compared to the MEND-simulated mean R_h under control, changing soil temperature under warming would result in a 22.2% increase in R_h , whereas changing soil moisture would cause a decrease in R_h by 8.1% (Fig. S14). Although the negative effect on R_h due to slightly drier soil under warming was considerable, it was completely shifted by the significant positive effect from soil temperature increase”* (Line 319-325).

B3. Data availability

Second, an issue I also brought up in my previous review is that model output and respiration measurements do not seem to be available anywhere. This would be particularly helpful for evaluating and interpreting the results related to changing Q_{10} values. It would be helpful to have figures in the supplement showing the respiration vs temperature relationship and Q_{10} model fit line for each year in warmed and control treatments. Some of the Q_{10} fits are statistically significant but have low R^2 (Table S1), and being able to visualize those fits is important context for interpreting the results of the paper. Ideally, model output and respiration measurements would be available as raw data or spreadsheets (machine-readable formats) in the interest of open data and reproducibility. The current manuscript does not provide information on how many total respiration measurements there were or when they were taken beyond vague information (“measured once or twice a month”). For figures like Figure 3b, this makes it impossible to tell how many observations are being shown or how they were distributed in time. A table that included every respiration measurement and the date it was collected, along with soil temperature and moisture associated with the measurement, in a format that could be downloaded and analyzed, would solve this problem and support discoverability and reproducibility according to current scientific standards. “Available from the author upon request” is not a good solution to this issue.

Responses: Thanks for pointing these issues out. We agree that the additions of raw data and figures about model output and respiration measurements in the manuscript are helpful for evaluating and interpreting our results. Now we had added a new Fig. S3 in the supplement to show the respiration vs temperature relationship and Q_{10} model fit line for each year in control and warming treatments. Significant ($p < 0.05$) or marginally significant ($p < 0.10$) apparent Q_{10} estimates were observed under both control and warming treatments in all years except 2011. The low R^2 values for some of the Q_{10} were due to the extremely soil moisture limitation in the drought years, such as 2011 and 2012. Furthermore, a table including model output and every respiration measurement with soil temperature and moisture associated with the measurement was provide in an excel file. We put the sentence that “Available from the author upon request” was just followed some previous published papers for this type of data. Actually, all of our

molecular data are deposited in public databases by following general standards for data deposition.

B4.

Line 62: It is ambiguous what “whose” refers to in this sentence. This sentence should be reworded.

Responses: Thanks for pointing this out! We have rewritten this sentence in the revised manuscript as “Soil respiration is the largest single source of carbon dioxide (CO₂) from terrestrial ecosystems to the atmosphere, and is about ten times larger than anthropogenic emissions” (Line 61-62).

B5.

Line 110-111: “right after the continuous warming” needs rewording

Responses: This sentence have been reworded in our revised manuscript as “The warmed plots were subjected to continuous warming by infrared radiators (+3 °C), and annual soil samples were archived over subsequent years and analyzed by integrated metagenomics technologies” (Line 111-113).

B6.

Line 272: “preferred for better global C cycle modeling” could be edited to be more specific about why these numbers are preferred, i.e. they match better with observation-based estimates

Responses: We modified this statement in the revised manuscript as “In addition, the MEND-derived intrinsic Q_{10} values were confined from 1.20–2.42 (tMEND) to a narrow range of 1.27–2.13 (gMEND), corroborating that Q_{10} values of 2 or below are usually used in global C cycle modeling” (Line 338-340).

B7.

Line 275-276: The statement that using MEND can remove confounding effects of other environmental factors would be a good place to introduce the idea of using the model to estimate the contribution of changing soil moisture vs changing temperature to changes in respiration

Responses: We used MEND to estimate the R_h response to a single-factor change in soil temperature or moisture during the seven-year experimental period. Compared to the MEND-simulated mean R_h under the control treatment, changing soil temperature under warming treatment would result in a 22.2% increase in R_h , whereas changing soil moisture would cause a decrease in R_h by 8.1% (Fig. S14). Although the negative effect on R_h due to slightly drier soil under warming treatment was considerable, it was completely shifted by the significant positive

effect by increasing soil temperature. The corresponding changes have been made in Line 315-325.

B8.

Line 460: It's not clear how weighting factors were assigned

Responses: We modify the sentence to show how weighting factors were assigned. The corresponding changes have been made in the revised manuscript as “Because we have far more R_h observations (i.e., 74 in control or warmed cases) than the other variables and R_h is the most important variable in soil C studies, we assign a much higher weighting factor to R_h than the other three objective functions (MBC, EnzCo, and EnzCh), i.e. $w_1 = 5/8$ and $w_2 = w_3 = w_4 = 1/8$ ” (Line 530-533)

B9.

Line 753-757: The explanation of how the bars were calculated is difficult to understand and is pretty complex to be in a figure caption. Consider rewriting or simplifying.

Responses: The figure legends for Figure 2c have been rewritten in the revised manuscript as: “The relative proportion of significantly warming-stimulated and significantly warming-inhibited genes in biogeochemical cyclings according to GeoChip data. Dash line represents that the abundance of warming-stimulated (red) genes are equal to the abundance of warming-inhibited (blue) genes. Significance is based on response ratio of each gene with 95% confidence intervals. Biogeochemical cycling genes included all genes involved in C degradation, C fixation, N cycling, phosphorus (P) utilization and sulfur (S) metabolism” (Line 841-845).

REVIEWER COMMENTS

Reviewer #1 (Remarks to the Author):

The authors have adequately addressed most of my concerns, except for one important one. I believe it is time to apply more rigor to the use of the term "intrinsic sensitivity" as it is clearly being defined differently by various authors. This is more than a semantic issue, because it implies a property that should be relatable to a biological process.

In the years preceding my review paper with Ivan Janssens (Davidson & Janssens, 2006), there had been several reports of Q10s of respiration of 1 or less (no or negative temperature sensitivity) to >3 and even as high as 50. Based on a long tradition of assuming that biological processes usually have a Q10 of about 2, and backed up by the metabolic theory of ecology work by Brown, Gillooly, Enquist, and others (e.g., ref 12 of the present manuscript), which shows that most respiratory enzymatic processes have an activation energy of about 50-60 kJ/mol, we proposed that the intrinsic temperature sensitivity should be equal to the activation energy of the most common respiratory enzymes. However, the apparent temperature sensitivity of measured respiration could be lower if there are other factors that constrain enzymatic activity (e.g., substrate limitation, microbial desiccation) or could be higher if other factors are positively confounded with temperature (e.g. the phenology of GPP or the phase change of soil water with thawing). An activation energy of 50-60 kJ/mol is roughly equivalent to a Q10 of 2.0-2.5 in the temperature range of 5-25C.

The present manuscript cites Mahecha et al (ref 34) for its distinction between intrinsic and apparent temperature sensitivity, but, unfortunately, their derived intrinsic temperature sensitivity is very different from what D&J proposed in 2006, and it diverges from metabolic theory expectations of an intrinsic sensitivity of about $Q_{10} \approx 2$ or $E_a \approx 55$. They used a statistical approach to detrend the seasonality of Fluxnet datasets on total ecosystem respiration (TER), and they defined the Q10 of the seasonally detrended TER as their "intrinsic" temperature sensitivity, which was about 1.4. Detrending is important because respiration is driven by both temperature and substrate production from GPP, and the seasonal trends of those two factors are confounded (i.e., higher temperature and higher GPP during summer in temperate climates; see Curiel Yuste et al. 2004 GCB 10:161–169). However, the resulting seasonally detrended temperature sensitivity should have been labeled "detrended apparent temperature sensitivity" rather than "intrinsic sensitivity," because, as Mahecha et al acknowledged, there were still other factors at the ecosystem scale that likely limited the expression of the intrinsic temperature sensitivity of enzymatic activity: "However, given the nontrivial ecophysiological interpretation of a multitude of processes summing up to the observed ecosystem respiration, our results do not justify the prescription of $Q_{10} = 1.4$ for all rate constants in soil carbon models. Rather, a deeper understanding of the different factors and processes limiting soil carbon metabolization is needed ..." Nutrients and water (both desiccation stress effects on diffusion of substrates in water films) can also suppress the expression of the intrinsic temperature sensitivity of enzymes.

The present authors also cite the work of Zhou et al. (ref 33) as an example of calculating an intrinsic temperature sensitivity based on models that aspire to separate out the effects of temperature from the effects of other potentially confounding factors, such as soil moisture, thereby allowing the fit of a Q10 parameter that would reflect the intrinsic temperature sensitivity of the dominant enzymes without constraints of substrate supply or other moisture effects. Zhou et al. used a the 5-pool TECO model, whereas the current manuscript applies the same approach to the more complex MEND model, which includes microbial processes, among other soil C stabilization processes. In theory, this approach is valid, but one should do a reality check to see if the derived "intrinsic" Q10 values make sense as being representative of enzymatic processes. In Zhou et al. they range from about 1.4 to 2.5, which would be an E_a of about 25-65 kJ/mol in the temperature range of 10-30C. The upper end of that Q10/ E_a range is reasonable, but the lower end of the range is not, so one might deduce that something is either missing in the model or that

some other parameters are not fitted ideally. Similarly, the Q10 value of 1.39 reported for "intrinsic" temperature sensitivity in the present manuscript would equate to an E_a of about 25 kJ/mol, which is unrealistically low for the most common respiration enzymes, even with the most extreme thermal adaptation imaginable. There are some enzymes with E_a 's that low, but they could not be the dominant ones producing most of the heterotrophic production of CO₂. This suggest a model failure of some sort, from which, perhaps, we could learn something. Indeed, we tend to learn more when models fail than when they work. I see in Tables S5 and S9 that carbon use efficiency (CUE) is an important part of the MEND model, and that the CUE has an assumed temperature sensitivity. There is not a strong consensus in the literature whether CUE is temperature dependent or even what the sign of the temperature dependency of CUE should be (see Hagerty et al. Biogeochemistry 2018, 140:269–283, for a discussion of the uncertainties of the temperature sensitivity of CUE and the importance of such assumptions for C model simulations). How sensitive is the inferred "intrinsic" sensitivity of the MEND model to its assumed temperature sensitivity of CUE? I don't have a deep understanding of the MEND model, so it could well be that some other model processes, other than or in addition to CUE, somehow interact with temperature-dependent functions with uncertain assumptions and parameterizations (Eqs E4-6 in Table S5 are apparently applied to other unspecified temperature response functions). In any case, I would argue that because the inferred "intrinsic" temperature sensitivity ($Q_{10} = 1.39$) requires an enzymatic E_a as low as 25 kJ/mol, which is unlikely, then there must be some other problem with the model structure or parameterization. The authors specifically state that "the Q10 in the MEND model solely reflects the microbial and enzymatic responses to temperature change." However, their interpretation of a Q10 of 1.39 is not in alignment with a large literature on the intrinsic E_a 's of common respiratory enzymes. Production of isoenzymes with lower E_a 's is a plausible thermal adaptation response, but halving the E_a seems unlikely. Note that once synthesized, an enzyme's E_a does not change, although the activity of the enzyme can be suppressed by factors other than temperature. The inferred MEND-model temperature sensitivity of $Q_{10} = 1.77$ ($E_a \approx 40$ kJ/mol) in the control plots is also unrealistically low, suggesting that there are also factors such as periods of suboptimal soil moisture or substrate supply that suppress the expected intrinsic temperature sensitivity ($E_a \approx 50$ -60 kJ/mol) of the enzymatic processes of respiration under the control treatment, and that these limitations do not appear to be fully accounted for by the model.

The authors may choose to investigate further their model structure, parameterization, and sensitivity to assumptions. They may choose to rename and reframe what the MEND-derived Q10 represents. This work demonstrates that a variety of potential thermal adaptation responses, including changes in microbial community composition, can lower the overall temperature sensitivity of the integrated decomposition process, but lowering the E_a of the dominant enzymes from the expected 50-60 kJ/mol to 25 kJ/mol is an unlikely explanation.

I hope that these comments are helpful and that this discussion can lead to advancement of our understanding of the processes that we are modeling and the meaning of the model parameters.

Reviewer #2 (Remarks to the Author):

After reading through the revised manuscript and response to reviewers, I am satisfied that the authors have addressed the key points brought up in the reviews. The addition of the data table in supplemental information and the discussion of temperature versus moisture effects in MEND simulations are helpful improvements. The temperature versus Rh plots that were added to supplemental material are very helpful context for the results.

I have a few suggestions that may improve the clarity of some of these new results, but overall I

do not see any remaining issues that need to be fixed.

1. Role of moisture: The comparative responses of MEND simulations to warming and drying in the experiment provide very useful context to the results. I think the conclusion in lines 323-325 could be clarified. While it is true that the temperature effect outweighed the moisture effect, the focus of the manuscript is on thermal adaptation, i.e. a relative reduction of the strength of the temperature fund. In fact, the magnitude of the moisture effect (8%) was very close to the magnitude of the thermal adaptation effect (11%). Since the intrinsic thermal adaptation was calculated using MEND simulations that incorporated both temperature and moisture effects, I think that the thermal adaptation result takes moisture effect into account. But I think it would be helpful to state this explicitly rather than dismissing the moisture effect as less than the total warming effect.

2. Observed range of temperatures: The new temperature versus Rh plots in Figure S3 were a very helpful addition to the manuscript. One aspect of the results that these figures show, which was not obvious previously, is that the temperature in some years reached very high values (30-40 C). The highest temperatures were reached in 2011 and 2012, which were also years with the lowest apparent Q10 and the lowest correlation between temperature and Rh. I think it is worth acknowledging in the text that a monotonic, increasing function like Q10 may be a poor fit for higher temperatures that may be beyond the optimal temperature for microbial respiration. In these cases, respiration might be suppressed rather than enhanced by the microbial response to increasing temperatures. See, for example, Alster et al., 2018 (<http://doi.wiley.com/10.1111/gcb.14342>), which found a mean optimum temperature for microbial respiration of 29 C.

Other comments:

Line 197: At this point I think it makes more sense to describe the result as a decrease in temperature sensitivity rather than thermal adaptation. This would make it more consistent with the next paragraph, which discusses the decrease in temperature sensitivity and the different potential causes for it (which include thermal adaptation).

Line 250: I would include a reference to Fig 2b when mentioning the SEM analysis.

Line 863/Figure 4: The caption here says that the reduction of Rh due to thermal adaptation was 8.2%, while in the text (line 348) it says that thermal adaptation would reduce Rh by 11.6%. But using the numbers in the figure, $(2.18 - 2.03)/2.18 = 6.9\%$. Shouldn't these numbers match? I suggest double checking these numbers.

Figure S5: Do the BIOLOG results have units associated with them?

Figure S13 and S14: Specify which version of MEND (tMEND or gMEND) is shown

Reviewer #3 (Remarks to the Author):

The manuscript entitled "Soil Carbon Loss Due to Persistent Microbial Adaptation to Climate Warming" is an impressive work of experimentation and modeling. The authors clearly have spent a lot of time and effort crafting this interesting manuscript. However, I have to agree with Reviewer #1 about their concerns with how temperature sensitivity is characterized. There is much evidence that Q10 is a very poor indicator of temperature sensitivity in soil systems. Therefore, in my opinion, this manuscript is not "cutting edge soil microbial ecology," at least from a temperature sensitivity perspective. Since Q10 is fundamentally dependent on the temperatures that it's measured at it is impossible to extrapolate any meaning from the Q10 values. If they were to estimate Q10 at a different range of values the responses would be very different as you see in Fig S3. This is an artifact of the data fitting, not necessarily indicative of the "true" temperature response (see Sierra 2012 Biogeochem). Therefore, whether or not the "intrinsic" versus "apparent" Q10 values make sense is arbitrary.

Here are some possible ideas for modifying the manuscript:

- 1) Can they show that their temperature data is clearly monotonic and follows an Arrhenius approach? Looks like this could be true for some years (Fig S3). If this is the case, then microbes/enzymes have virtually no impact on the temperature response (Schipper et al., 2019 Ag, Eco & Env).
- 2) Can they amend their model to use a non Q10 or Arrhenius metric for temperature sensitivity? See Schipper et al., 2014 GCB or even Baath 2018 GCB for some ideas.

In regard to the use of the term "intrinsic," I would recommend just changing this term to avoid confusion. While I acknowledge that they are using "intrinsic" and "apparent" temperature sensitivity as was published in Zhou 2018, the Davidson and Janssens 2018 Nature paper identifying these terms in a different way is so iconic that I think it is confusing to use the same terms with a different definition. Instead, can the authors change the terms to something like "microbial" response of temperature or "model-derived" temperature response? I think this subtle change would make the manuscript clearer and appease Reviewer #1.

Responses to Comments from Reviewers

We have addressed the reviewers' comments and questions point-by-point. The original reviewer's comments are italicized and our responses to the reviewer's comments follow. The numbers of lines in the text are referred to the revised version, where corrections are tracked.

A. Responses to Reviewer #1 (Remarks to the Author):

A1. intrinsic temperature sensitivity

The authors have adequately addressed most of my concerns, except for one important one. I believe it is time to apply more rigor to the use of the term “intrinsic sensitivity” as it is clearly being defined differently by various authors. This is more than a semantic issue, because it implies a property that should be relatable to a biological process.

Responses: We agree with the reviewer that the Q_{10} estimated from ecosystem modeling could not represent the “intrinsic temperature sensitivity” as proposed by Davidson & Janssens (2006). Therefore, we have changed the term from “intrinsic temperature sensitivity” to “model-derived temperature sensitivity” throughout the manuscript.

A2. Model-derived Q_{10} values

In the years preceding my review paper with Ivan Janssens (Davidson & Janssens, 2006), there had been several reports of Q_{10} s of respiration of 1 or less (no or negative temperature sensitivity) to >3 and even as high as 50. Based on a long tradition of assuming that biological processes usually have a Q_{10} of about 2, and backed up by the metabolic theory of ecology work by Brown, Gillooly, Enquist, and others (e.g., ref 12 of the present manuscript), which shows that most respiratory enzymatic processes have an activation energy of about 50-60 kJ/mol, we proposed that the intrinsic temperature sensitivity should be equal to the activation energy of the most common respiratory enzymes. However, the apparent temperature sensitivity of measured respiration could be lower if there are other factors that constrain enzymatic activity (e.g., substrate limitation, microbial desiccation) or could be higher if other factors are positively confounded with temperature (e.g. the phenology of GPP or the phase change of soil water with thawing). An activation energy of 50-60 kJ/mol is roughly equivalent to a Q_{10} of 2.0-2.5 in the temperature range of 5-25C.

Responses: Thanks for the reviewer's insightful comments on the values of Q_{10} and activation energy (Ea). We compiled a dataset of 96 Ea values (33 for ligninases and 63 for cellulases) from

around 60 publications (See Table S11 and Fig. S15, also shown below). The E_a values for ligninases and cellulases are 48 ± 18 and 39 ± 15 kJ mol^{-1} , respectively. By combining the literature data for both ligninases and cellulases, we show that $E_a = 42 \pm 17$ (mean \pm standard deviation) kJ mol^{-1} , with a 95% confidence interval (95% CI) of 14–79 kJ mol^{-1} . The corresponding literature- $Q_{10} = 1.82 \pm 0.44$ (95% CI = 1.21–2.93) with a temperature increase from 20 °C to 30 °C.

The MEND-derived $Q_{10} = 1.77 \pm 0.12$ for control and 1.39 ± 0.09 for warming. We also derived Q_{10} by the TECO model: 1.79 ± 0.09 (control) and 1.50 ± 0.15 (warming). The Q_{10} results under control were very close between two models but higher from TECO than from MEND under warming.

Fig. S15 Activation energy (E_a) and corresponding Q_{10} values from literature and our model estimates. Literature- E_a values are pooled data from major ligninases and cellulases catalyzing the decomposition of soil organic carbon. Literature- Q_{10} values are calculated from E_a with a temperature increase from 20 °C to 30 °C. Model-derived Q_{10} values are those under control and warming (+3°C) treatments. Model- E_a values are calculated from Q_{10} with a temperature increase from 20 °C to 30 °C.

As per the control treatment, the MEND-derived mean Q_{10} values were slightly lower (2.7%) than the literature mean Q_{10} (1.77 vs. 1.82). However, the model-derived mean Q_{10} values under warming were at the lower bound of one standard deviation (1.39 vs. 1.38).

In conclusion, we agree with the reviewer that our model-derived Q_{10} and E_a under warming treatment may not represent the intrinsic temperature sensitivity of the most respiratory enzymes. This discrepancy could be attributed to two reasons: (i) the lack of a thorough representation of

all factors confounded with temperature; and (ii) the uncertainty in multiple model parameters that were not constrained ideally.

While we realize that no model is perfect, the relative difference in model-derived Q_{10} between control and warming potentially illustrated the decreased temperature sensitivity under warming treatment based on the consistent model parameterization procedure given current model assumptions.

A3. Comparison with publications by Mahecha et al. (2010) and Zhou et al. (2018)

The present manuscript cites Mahecha et al (ref 34) for its distinction between intrinsic and apparent temperature sensitivity, but, unfortunately, their derived intrinsic temperature sensitivity is very different from what D&J proposed in 2006, and it diverges from metabolic theory expectations of an intrinsic sensitivity of about $Q_{10} \approx 2$ or $E_a \approx 55$. They used a statistical approach to detrend the seasonality of Fluxnet datasets on total ecosystem respiration (TER), and they defined the Q_{10} of the seasonally detrended TER as their “intrinsic” temperature sensitivity, which was about 1.4. Detrending is important because respiration is driven by both temperature and substrate production from GPP, and the seasonal trends of those two factors are confounded (i.e., higher temperature and higher GPP during summer in temperate climates; see Curiel Yuste et al. 2004 GCB 10:161–169). However, the resulting seasonally detrended temperature sensitivity should have been labeled “detrended apparent temperature sensitivity” rather than “intrinsic sensitivity,” because, as Mahecha et al acknowledged, there were still other factors at the ecosystem scale that likely limited the expression of the intrinsic temperature sensitivity of enzymatic activity: “However, given the nontrivial ecophysiological interpretation of a multitude of processes summing up to the observed ecosystem respiration, our results do not justify the prescription of $Q_{10} = 1.4$ for all rate constants in soil carbon models. Rather, a deeper understanding of the different factors and processes limiting soil carbon metabolization is needed ...” Nutrients and water (both desiccation stress effects on diffusion of substrates in water films) can also suppress the expression of the intrinsic temperature sensitivity of enzymes.

The present authors also cite the work of Zhou et al. (ref 33) as an example of calculating an intrinsic temperature sensitivity based on models that aspire to separate out the effects of temperature from the effects of other potentially confounding factors, such as soil moisture, thereby allowing the fit of a Q_{10} parameter that would reflect the intrinsic temperature sensitivity of the dominant enzymes without constraints of substrate supply or other moisture effects. Zhou et al. used a the 5-pool TECO model, whereas the current manuscript applies the same approach to the more complex MEND model, which includes microbial processes, among other soil C stabilization processes. In theory, this approach is valid, but one should do a reality check to see if the derived “intrinsic” Q_{10} values make sense as being representative of enzymatic processes. In Zhou et al. they range from about 1.4 to 2.5, which would be an E_a of about 25-65 kJ/mol in

the temperature range of 10-30C. The upper end of that Q_{10}/E_a range is reasonable, but the lower end of the range is not, so one might deduce that something is either missing in the model or that some other parameters are not fitted ideally. Similarly, the Q_{10} value of 1.39 reported for "intrinsic" temperature sensitivity in the present manuscript would equate to an E_a of about 25 kJ/mol, which is unrealistically low for the most common respiration enzymes, even with the most extreme thermal adaptation imaginable. There are some enzymes with E_a 's that low, but they could not be the dominant ones producing most of the heterotrophic production of CO₂. This suggest a model failure of some sort, from which, perhaps, we could learn something. Indeed, we tend to learn more when models fail than when they work.

Responses: Thanks for pointing out that the Q_{10} estimated by Mahecha et al. (2010) refers to “detrended apparent temperature sensitivity”.

The model-derived Q_{10} values in this study are more like the ones reported by Zhou et al. (2018). Actually, we used both TECO and MEND models to derive Q_{10} in this study and the results from both models are close to each other (see Responses A2).

We agree with the reviewer that these model-derived Q_{10} are not necessarily an indicative of the intrinsic Q_{10} , although we have attempted to separate the effects of temperature from the effects of other potentially confounding factors, such as soil moisture, as “*there were still other factors at the ecosystem scale that likely limited the expression of the intrinsic temperature sensitivity of enzymatic activity*” (Line 389-391).

A4. How sensitive is the model-derived Q_{10} to assumed temperature sensitivity of CUE

I see in Tables S5 and S9 that carbon use efficiency (CUE) is an important part of the MEND model, and that the CUE has an assumed temperature sensitivity. There is not a strong consensus in the literature whether CUE is temperature dependent or even what the sign of the temperature dependency of CUE should be (see Hagerty et al. Biogeochemistry 2018, 140:269–283, for a discussion of the uncertainties of the temperature sensitivity of CUE and the importance of such assumptions for C model simulations). How sensitive is the inferred "intrinsic" sensitivity of the MEND model to its assumed temperature sensitivity of CUE?

Responses: Thanks for the insightful comments. We acknowledge that “there is not a strong consensus in the literature whether CUE is temperature dependent or even what the sign of the temperature dependency of CUE should be”. For example, CUE decreases with increasing temperature based on laboratory studies (DeVêvre & Horwáth, 2000; Fieschko & Humphrey, 1984; Frey, Lee, Melillo, & Six, 2013; Steinweg, Plante, Conant, Paul, & Tanaka, 2008; Tucker, Bell, Pendall, & Ogle, 2013). However, a positive dependence of CUE on site level mean annual temperature (MAT) could capture the soil respiration patterns across biomes (Sinsabaugh, Moorhead, Xu, & Litvak, 2017; Xu et al., 2017; J.-S. Ye, Bradford, Maestre, Li, & García-

Palacios, 2020; J. S. Ye, Bradford, Dacal, Maestre, & Garca-Palacios, 2019). Although these studies are advantageous in addressing spatial pattern of the temperature sensitivity of soil respiratory loss given the quasi steady state assumption, more effort will be needed to elucidate whether projections of temporal patterns of key soil variables (soil respiration and soil organic carbon) are robust, as the temperature sensitivity of soil organic matters may not be directly inferred from spatial gradients (Abramoff, Torn, Georgiou, Tang, & Riley, 2019). In addition, the mismatch between spatiotemporal scales could occur when the site level MAT is correlated to *CUE* values, because these *CUE* estimates were generally estimated from various short-term laboratory experiments, which, based on our study, might be overestimated when compared with those based on long-term experiments (Li et al., 2019).

The divergent *CUE* estimates might be due to different definitions and quantification methods of the differential microbial mechanisms operating at contrasting spatiotemporal scales (Geyer, Dijkstra, Sinsabaugh, & Frey, 2019), e.g., day vs. decade, population vs. community vs. ecosystem (Geyer, Kyker-Snowman, Grandy, & Frey, 2016), and temporal vs. spatial gradient (Abramoff et al., 2019; Sinsabaugh et al., 2017).

In the MEND model, we define a parameter (true growth yield: Y_g) to separate microbial growth from growth respiration (Wang & Post, 2012). The MEND model also simulates microbial maintenance, mortality, and enzyme production. Thus, the apparent *CUE*, different from the parameter Y_g , is **NOT** a parameter in the MEND model. On the contrary, the apparent *CUE* in MEND is represented by considering explicit microbial processes as suggested by Hagerty et al. (Hagerty, Allison, & Schimel, 2018)

Even the parameter Y_g will change with temperature in the MEND model:

$$Y_g(T) = Y_g(T_{ref}) - k_{Yg} \cdot (T - T_{ref})$$

where $Y_g(T)$ and $Y_g(T_{ref})$ are the Y_g at soil temperature T and T_{ref} (reference temperature), respectively; and k_{Yg} denote the temperature sensitivity of Y_g .

To test the temperature dependency of $Y_g(T)$, we had estimated the slope ($-k_{Yg}$) using 22-year of warming experimental data at the Harvard Forest (Li et al., 2019). Although the slope $-k_{Yg}$ had a wide *a priori* range from -0.017 to 0.017 , the posterior range of $-k_{Yg}$ was shrunk to -0.01 ± 0.005 and $-k_{Yg}$ was significantly less than 0. Therefore, the range of $-k_{Yg}$ was set to between -0.016 and -0.001 in this study. Our results show that Q_{10} and k_{Yg} were NOT correlated to each other (correlation coefficient = 0.0016, p-value = 0.94, see below Fig. S16), which indicates that the MEND-derived Q_{10} was NOT sensitive to its assumed temperature sensitivity of Y_g .

Fig. S16 Correlation between Q_{10} and k_{Y_g} (temperature sensitivity of Y_g). Y_g is the true growth yield, e.g., a proxy for carbon use efficiency (CUE) in the MEND model. The temperature dependence of Y_g on soil temperature (T) is described by $Y_g(T) = Y_g(T_{ref}) - k_{Y_g} \cdot (T - T_{ref})$, where $Y_g(T)$ and $Y_g(T_{ref})$ are the Y_g at soil temperature T and T_{ref} (reference temperature), respectively; and k_{Y_g} denote the temperature sensitivity of Y_g .

A5. Other factors suppress the temperature sensitivity

I don't have a deep understanding of the MEND model, so it could well be that some other model processes, other than or in addition to CUE, somehow interact with temperature-dependent functions with uncertain assumptions and parameterizations (Eqs E4-6 in Table S5 are apparently applied to other unspecified temperature response functions). In any case, I would argue that because the inferred "intrinsic" temperature sensitivity ($Q_{10} = 1.39$) requires an enzymatic E_a as low as 25 kJ/mol, which is unlikely, then there must be some other problem with the model structure or parameterization. The authors specifically state that "the Q_{10} in the MEND model solely reflects the microbial and enzymatic responses to temperature change." However, their interpretation of a Q_{10} of 1.39 is not in alignment with a large literature on the intrinsic E_a 's of common respiratory enzymes. Production of isoenzymes with lower E_a 's is a plausible thermal adaptation response, but halving the E_a seems unlikely. Note that once

synthesized, an enzyme's E_a does not change, although the activity of the enzyme can be suppressed by factors other than temperature. The inferred MEND-model temperature sensitivity of $Q_{10} = 1.77$ ($E_a \approx 40$ kJ/mol) in the control plots is also unrealistically low, suggesting that there are also factors such as periods of suboptimal soil moisture or substrate supply that suppress the expected intrinsic temperature sensitivity ($E_a \approx 50$ -60 kJ/mol) of the enzymatic processes of respiration under the control treatment, and that these limitations do not appear to be fully accounted for by the model.

The authors may choose to investigate further their model structure, parameterization, and sensitivity to assumptions. They may choose to rename and reframe what the MEND-derived Q_{10} represents. This work demonstrates that a variety of potential thermal adaptation responses, including changes in microbial community composition, can lower the overall temperature sensitivity of the integrated decomposition process, but lowering the E_a of the dominant enzymes from the expected 50-60 kJ/mol to 25 kJ/mol is an unlikely explanation.

I hope that these comments are helpful and that this discussion can lead to advancement of our understanding of the processes that we are modeling and the meaning of the model parameters.

Responses: We appreciate the constructive comments and suggestions. Here we summarize the key points based on our detailed responses A1–A4:

- (1) We have changed the term from “intrinsic temperature sensitivity” to “model-derived temperature sensitivity” throughout the manuscript.
- (2) The model-derived temperature sensitivity are not necessarily an indicative of the intrinsic temperature sensitivity, although we have attempted to separate the effects of temperature from the effects of other potentially confounding factors, such as soil moisture, as “*there were still other factors at the ecosystem scale that likely limited the expression of the intrinsic temperature sensitivity of enzymatic activity*” (Line 389–391). In addition, limitations in model structure and model parameterization in terms of parameter uncertainty could hinder a thorough differentiation between confounding factors as well as between intrinsic and apparent temperature sensitivity.
- (3) The MEND-derived Q_{10} was NOT sensitive to its assumed temperature sensitivity of Y_g .
- (4) The MEND-derived mean Q_{10} value under control was only 2.7% lower than the literature mean Q_{10} (1.77 vs. 1.82). While we acknowledge that our model-derived Q_{10} under warming was at the lower bound of one standard deviation (1.39 vs. 1.38), the relative difference in model-derived Q_{10} between control and warming potentially illustrated the decrease in temperature sensitivity under warming conditions based on the consistent model parameterization procedure given current model assumptions.

To address all these concerns (A1–A5) from Reviewer 1, we added the following sentences in text “The MEND-derived Q_{10} values (1.77 ± 0.12 for control, and 1.39 ± 0.09 for warming) were close to those estimated from the TECO model in the current study (1.79 ± 0.09 for control, and 1.50 ± 0.15 for warming), as well as a previous study with the TECO model (1.4–2.5) (Zhou et al., 2018). Our model-derived Q_{10} under warming was similar to the detrended temperature sensitivity (1.4 ± 0.1) estimated across 60 FLUXNET sites (Mahecha et al., 2010). We also compared our results to a meta-analysis of activation energy (E_a) and Q_{10} values for cellulases and ligninases from ca. 60 publications (see Table S11 and Fig. S15). The MEND-derived mean Q_{10} values under the control treatment were slightly (e.g., 2.7%) lower than the mean Q_{10} of those studies (1.77 vs. 1.82). However, the model-derived mean Q_{10} value under warming was at the lower bound of one standard deviation (1.39 vs. 1.38). We acknowledge that most C-degrading enzymatic processes have an activation energy of about 50–60 kJ mol⁻¹ (roughly equivalent to a Q_{10} of 2.0–2.5) (Davidson & Janssens, 2006). Therefore, the model-derived Q_{10} values may fail to catch the intrinsic temperature sensitivity of microbial and enzyme activities, although we have attempted to separate the effects of temperature from those of other potential confounding factors (e.g., soil moisture) through the process-based modeling. There are still other factors at the ecosystem level that likely limited the expression of the intrinsic temperature sensitivity of enzyme activity (Davidson & Janssens, 2006), which needs further research in future studies. In addition, limitations and uncertainties in model structure and parameterization could further hinder a thorough differential representation of the effects of multiple confounding factors (e.g., soil temperature and moisture, substrate supply and litter quality) on enzyme activities and microbial carbon use efficiency (CUE), though our results showed no significant correlation between Q_{10} and the temperature sensitivity of CUE (Fig. S16). Despite that more effort should be devoted to improving the representation of multi-factor effects on soil respiration processes as well as confining the uncertainties in model structure, parameterization, and input data, microbially-enabled ecosystem modeling renders a significant advance in our understanding of microbial responses to the changes in temperature” (Line 376–400).

B. Responses to Reviewer #2 (Remarks to the Author):

B1. Overall comments

After reading through the revised manuscript and response to reviewers, I am satisfied that the authors have addressed the key points brought up in the reviews. The addition of the data table in supplemental information and the discussion of temperature versus moisture effects in MEND simulations are helpful improvements. The temperature versus Rh plots that were added to supplemental material are very helpful context for the results.

I have a few suggestions that may improve the clarity of some of these new results, but overall I do not see any remaining issues that need to be fixed.

Response: Thank you very much for the complimentary comments and constructive suggestions.

B2. Role of moisture

Role of moisture: The comparative responses of MEND simulations to warming and drying in the experiment provide very useful context to the results. I think the conclusion in lines 323-325 could be clarified. While it is true that the temperature effect outweighed the moisture effect, the focus of the manuscript is on thermal adaptation, i.e. a relative reduction of the strength of the temperature fund. In fact, the magnitude of the moisture effect (8%) was very close to the magnitude of the thermal adaptation effect (11%). Since the intrinsic thermal adaptation was calculated using MEND simulations that incorporated both temperature and moisture effects, I think that the thermal adaptation result takes moisture effect into account. But I think it would be helpful to state this explicitly rather than dismissing the moisture effect as less than the total warming effect.

Responses: Thanks for pointing this out! We incorporated the reviewer's statement into the revised manuscript: "Compared to the MEND-simulated mean Rh under control, changing soil temperature under warming would result in a 22.2% increase in Rh, whereas changing soil moisture would cause a decrease in Rh by 8.1% (Fig. S14). Therefore, both temperature and moisture effects greatly contribute the MEND-derived thermal adaptation effect, as both of them were taken into account in MEND simulations" (Line 348-352).

B3. Observed range of temperature

Observed range of temperatures: The new temperature versus Rh plots in Figure S3 were a very helpful addition to the manuscript. One aspect of the results that these figures show, which was not obvious previously, is that the temperature in some years reached very high values (30-40 C).

The highest temperatures were reached in 2011 and 2012, which were also years with the lowest apparent Q_{10} and the lowest correlation between temperature and Rh. I think it is worth acknowledging in the text that a monotonic, increasing function like Q_{10} may be a poor fit for higher temperatures that may be beyond the optimal temperature for microbial respiration. In these cases, respiration might be suppressed rather than enhanced by the microbial response to increasing temperatures. See, for example, Alster et al., 2018 (<http://doi.wiley.com/10.1111/gcb.14342>), which found a mean optimum temperature for microbial respiration of 29 C.

Responses: Thanks for your suggestion. We agree that apparent Q_{10} is a poor fit for some cases (e.g., in the year of 2011 and 2012) that may be due to (i) the confounding effects of environmental factors other than temperature and (ii) soil temperature beyond the optimal for microbial respiration. We have carefully read the articles you recommended, and agree that the temperatures (30-40 °C) may be beyond the mean optimum temperature for microbial respiration. Therefore, we add the following sentence in the revised manuscript: “*The poor fit of apparent Q_{10} in 2011 and 2012 is most likely due to the suppression rather than enhancement of microbial respiration under warming, which could be explained by the higher temperatures (e.g., >30 °C) beyond the optimal temperature for microbial respiration (Alster, Weller, & von Fischer, 2018) and/or the confounding effects of environmental factors other than temperature (e.g., soil moisture)*” (Line 195-201).

B4. Decrease in temperature sensitivity rather than thermal adaption

Line 197: At this point I think it makes more sense to describe the result as a decrease in temperature sensitivity rather than thermal adaptation. This would make it more consistent with the next paragraph, which discusses the decrease in temperature sensitivity and the different potential causes for it (which include thermal adaptation).

Responses: Thanks for pointing this out! We have rewritten this sentence in the revised manuscript: “*Altogether, the above results indicate that there was a strong and persistent decrease in temperature sensitivity of microbial heterotrophic respiration under warming over the last 7 years*” (Line 221-223).

B5. Reference for the SEM analysis

Line 250: I would include a reference to Fig 2b when mentioning the SEM analysis.

Responses: A reference to Fig. 2b has been added in the revised manuscript (Line 277).

B6. Reduction of Rh due to thermal adaption

Line 863/Figure 4: The caption here says that the reduction of Rh due to thermal adaptation was

8.2%, while in the text (line 348) it says that thermal adaptation would reduce R_h by 11.6%. But using the numbers in the figure, $(2.18 - 2.03)/2.18 = 6.9\%$. Shouldn't these numbers match? I suggest double checking these numbers.

Responses: Sorry for the confusion. Due to the uncertainties in the model-derived Q_{10} under control (1.77 ± 0.09) and under warming (1.39 ± 0.09), and to make the results more meaningful, we decided to use the mean heterotrophic respiration (R_h) in the control plot (R_h^{CT}), which represents the R_h in real ambient condition, as a consistent baseline to quantify the percent reduction in R_h (see Methods Line 638–645). We calculated the thermal adaptation effect as the percent reduction in R_h due to thermal adaptation relative to the baseline R_h , i.e, the mean R_h in the control plot (R_h^{CT})

$$\% \Delta R_h = (R_h^{woA} - R_h^{wA}) / R_h^{CT} \times 100\% = \Delta R_h^{woA-wA} / R_h^{CT} \times 100\%$$

$$\text{Using the numbers in Fig. 4a, } \% \Delta R_h = \frac{2.18 - 2.03}{1.84} = 8.2\%$$

Noting that Fig. 4a is a demonstration showing how we calculate the thermal adaption effect using the mean Q_{10} (1.77 for w/o adaption and 1.39 for with adaption under warming). However, Fig. 4b shows the uncertainty in the results due to the uncertainty in Q_{10} , and the mean value of all $\% \Delta R_h = 11.6\%$, which could be different from the specific $\% \Delta R_h = 8.2\%$ calculated from the mean Q_{10} values. Both of these $\% \Delta R_h$ values (11.6% and 8.2%) are within the uncertainty range of $\% \Delta R_h$ shown in Fig. 4b.

We have clarified this in the revised Fig. 4 caption (Line 925-938).

B7. BIOLOG units

Figure S5: Do the BIOLOG results have units associated with them?

Responses: BIOLOG MicroPlates are 96-well plates that contain pre-dried carbon sources and a tetrazolium violet redox dye. When the microbes can utilize the carbon source, the respiration of the microbial cells reduces the dye and the formation of purple color occurs. The color development is recorded as optical density (OD) at 590 nm. Therefore, BIOLOG results have no units associated with them.

B8. Version of MEND

Figure S13 and S14: Specify which version of MEND (tMEND or gMEND) is shown

Responses: All these MEND results refer to the gMEND. We have revised Figs. S13 and S14.

C. Responses to Reviewer #3 (Remarks to the Author):

C1. Q_{10} as indicator of temperature sensitivity

The manuscript entitled “Soil Carbon Loss Due to Persistent Microbial Adaptation to Climate Warming” is an impressive work of experimentation and modeling. The authors clearly have spent a lot of time and effort crafting this interesting manuscript. However, I have to agree with Reviewer #1 about their concerns with how temperature sensitivity is characterized. There is much evidence that Q_{10} is a very poor indicator of temperature sensitivity in soil systems. Therefore, in my opinion, this manuscript is not “cutting edge soil microbial ecology,” at least from a temperature sensitivity perspective. Since Q_{10} is fundamentally dependent on the temperatures that it's measured at it is impossible to extrapolate any meaning from the Q_{10} values. If they were to estimate Q_{10} at a different range of values the responses would be very different as you see in Fig S3. This is an artifact of the data fitting, not necessarily indicative of the “true” temperature response (see Sierra 2012 Biogeochem). Therefore, whether or not the “intrinsic” versus “apparent” Q_{10} values make sense is arbitrary.

Responses: We understand the reviewer’s concern. We agree that the Q_{10} method is empirical and imperfect and Q_{10} is dependent on the temperature (Davidson & Janssens, 2006; Sierra, 2012; Wang, Post, Mayes, Frerichs, & Jagadamma, 2012). We also agree that our apparent Q_{10} from curve-fitting and model-derived Q_{10} by process-based modeling are not necessarily indicative of the “true” temperature response, as it’s challenging to make a thorough exploration of the “true” temperature sensitivity. However, Sierra’s (2012) “analysis of the available empirical evidence shows that most studies actually agree with the Arrhenius and thermodynamics theory”, which doesn’t oppose the use of Q_{10} and activation-energy-based Arrhenius methods. In addition, our model-derived Q_{10} is estimated by process-based modeling that accounts for the effects of multiple factors, it is possible to address the temperature sensitivity of the soil decomposition and respiration processes according to Sierra (2012).

Here are some possible ideas for modifying the manuscript:

C2. Q_{10}

1) Can they show that their temperature data is clearly monotonic and follows an Arrhenius approach? Looks like this could be true for some years (Fig S3). If this is the case, then microbes/enzymes have virtually no impact on the temperature response (Schipper et al., 2019 Ag, Eco & Env).

Responses: We understand the reviewer’s concern. In Fig. S3, we show that the apparent relationship between R_h and soil temperature follow a monotonic exponential equation in most years. The poor fit of apparent Q_{10} in 2011 and 2012 indicates that the apparent temperature

sensitivity does not always follow a monotonic Arrhenius equation. However, the apparent temperature sensitivity estimate based on the field measurements are influenced by various other factors beyond temperature, including soil moisture, plants-derived substrate quality and availability, nutrient limitation influencing microbial enzyme production, experimental duration, and/or spatial heterogeneity, as well as uncertainty in instrumental measurements (Davidson & Janssens, 2006; Sierra, 2012). Therefore, the poor fit of Q_{10} in some cases doesn't necessarily imply that microbes/enzymes have virtually no impact on the temperature response.

To address this concern, we added the following sentences in the revised manuscript: “Significant ($p < 0.05$) or marginally significant ($p < 0.10$) apparent Q_{10} estimates were observed under both control and warming treatments in all years except 2011 (Fig. S3). Therefore, the apparent relationship between R_h and soil temperature follow a monotonic exponential equation in most years. The poor fit of apparent Q_{10} in 2011 and 2012 is most likely due to the suppression rather than enhancement of microbial respiration under warming, which could be explained by the higher temperatures (e.g., $>30^\circ\text{C}$) beyond the optimal temperature for microbial respiration (Alster et al., 2018) and/or the confounding effects of environmental factors other than temperature (e.g., soil moisture)” (Line 194-201).

C3. Use a non Q_{10} or Arrhenius metric for temperature sensitivity

2) Can they amend their model to use a non Q_{10} or Arrhenius metric for temperature sensitivity? See Schipper et al., 2014 GCB or even Baath 2018 GCB for some ideas.

Responses: We understand the reviewer's concern and acknowledge the reviewer's excellent suggestion. We could amend the MEND model to use a non Q_{10} or Arrhenius metric for temperature sensitivity. However, the application and evaluation of different temperature response functions is out of the scope of this study. Our objective is to compare the temperature sensitivity of soil heterotrophic respiration under ambient and warming treatment. We adopted the Q_{10} -based approach due to the following reasons:

- (1) The Q_{10} approach has been widely used and validated in the biological and environmental research (Davidson & Janssens, 2006). In addition, the temperature sensitivity of decomposition in most Earth System Models (ESMs) was described by the Q_{10} or Arrhenius equations (Todd-Brown et al., 2013).
- (2) The activation energy (E_a) in the Arrhenius equation can be converted to Q_{10} at given temperature (T) and the reference temperature (T_{ref}), and vice versa. This means we can interpret the Q_{10} temperature sensitivity by the Arrhenius's concept of activation energy (Davidson & Janssens, 2006).
- (3) In this study, the daily mean soil temperature generally ranged from -10°C to 30°C , with very few data (5%) between $30-40^\circ\text{C}$. The temperature values fall within the temperature range described by the Q_{10} or activation-energy-based Arrhenius relationship, which

generally holds over the temperature range of 0–40 °C (Brown, Gillooly, Allen, Savage, & West, 2004). It also indicates the soil temperature in this study generally below the optimum temperature (T_{opt}) as per the macromolecular rate theory (MMRT), e.g., $T_{opt} \geq 30$ °C for respiration and cellulase activity (Schipper, Hobbs, Rutledge, & Arcus, 2014) and $T_{opt} = 57$ – 71 °C reported by Schipper *et al.* (2019).

- (4) The square root equation proposed by Bååth (2018) could be another approach to investigate the temperature sensitivity of microbial growth and activity. However, if we want to elucidate the changes in temperature sensitivity and compare our results with previous studies, it is more convenient to convert the results to Q_{10} as shown by Bååth (2018).

In short, though the Q_{10} method is empirical and imperfect, it has been widely accepted to interpret the temperature sensitivity of the reaction rates in biological and chemical systems. In addition, the Q_{10} values have been well documented for soil biogeochemical reactions. All these enable us to examine and explain the temperature sensitivity in a generally accepted framework.

To address this concern, we added the following sentences in the revised manuscript: “*A wide range of different models have been developed to express the temperature sensitivity of SOM decomposition and respiration processes (Todd-Brown et al., 2013). While many models are based on the exponential function characterized by the Q_{10} or activation energy (Davidson & Janssens, 2006), the square root relationship (Bååth, 2018) and the macromolecular rate theory (MMRT) equation (Schipper et al., 2014) have also been proposed to enable the comparison of temperature sensitivity of microbial activity between habitats or organisms. The square root equation includes a theoretical minimum temperature for growth and activity, which allows to more accurately estimate Q_{10} below optimum temperature (Bååth, 2018). The core concept of the MMRT equation is that there exists an optimum temperature for enzyme and microbial activity (Schipper et al., 2014), which overcomes the limit of temperature range for the applicability of the Arrhenius and the square root equations. The optimum temperature in the MMRT equation could be ca. 30 °C (Alster et al., 2018; Schipper et al., 2014) and 57–71 °C (Schipper et al., 2019), which is generally above the temperature range of 0–40 °C for the validity of the Arrhenius relationship (Brown et al., 2004). Given that 95% of the soil temperatures were below 30 °C in our study site and the Q_{10} method has been widely accepted to interpret the temperature sensitivity in the biological and environmental research including most of the ESMS models (Davidson & Janssens, 2006; Todd-Brown et al., 2013), we adopted the Q_{10} approach (see Methods) to examine the apparent temperature sensitivity of microbial respiration (> 7 years) and their underlying mechanisms. This also allows us to directly compare our results to the vast amount of existing studies and interpret the temperature sensitivity in a generally accepted framework*” (Line 175-192).

C4. Intrinsic temperature sensitivity

In regard to the use of the term “intrinsic,” I would recommend just changing this term to avoid

confusion. While I acknowledge that they are using “intrinsic” and “apparent” temperature sensitivity as was published in Zhou 2018, the Davidson and Janssens 2018 Nature paper identifying these terms in a different way is so iconic that I think it is confusing to use the same terms with a different definition. Instead, can the authors change the terms to something like “microbial” response of temperature or “model-derived” temperature response? I think this subtle change would make the manuscript clearer and appease Reviewer #1.

Responses: Thank you for the constructive suggestion! We have changed the term from “intrinsic temperature sensitivity” to “model-derived temperature sensitivity” throughout the manuscript.

References:

- Abramoff, R. Z., Torn, M. S., Georgiou, K., Tang, J., & Riley, W. J. (2019). Soil Organic Matter Temperature Sensitivity Cannot be Directly Inferred From Spatial Gradients. *Global Biogeochemical Cycles*, 33(6), 761-776.
- Alster, C. J., Weller, Z. D., & von Fischer, J. C. (2018). A meta-analysis of temperature sensitivity as a microbial trait. *Global Change Biology*, 24(9), 4211-4224.
- Bååth, E. (2018). Temperature sensitivity of soil microbial activity modeled by the square root equation as a unifying model to differentiate between direct temperature effects and microbial community adaptation. *Global Change Biology*, 24(7), 2850-2861.
- Brown, J. H., Gillooly, J. F., Allen, A. P., Savage, V. M., & West, G. B. (2004). Toward a metabolic theory of ecology. *Ecology*, 85(7), 1771-1789.
- Davidson, E. A., & Janssens, I. A. (2006). Temperature sensitivity of soil carbon decomposition and feedbacks to climate change. *Nature*, 440(7081), 165.
- DeVêvre, O. C., & Horwáth, W. R. (2000). Decomposition of rice straw and microbial carbon use efficiency under different soil temperatures and moistures. *Soil Biology & Biochemistry*, 32(11-12), 1773-1785.
- Fieschko, J., & Humphrey, A. E. (1984). Statistical analysis in the estimation of maintenance and true growth yield coefficients. *Biotechnology and Bioengineering*, 26(4), 394-396.
- Frey, S. D., Lee, J., Melillo, J. M., & Six, J. (2013). The temperature response of soil microbial efficiency and its feedback to climate. *Nature Climate Change*, 3(4), 395-398.
- Geyer, K. M., Dijkstra, P., Sinsabaugh, R., & Frey, S. D. (2019). Clarifying the interpretation of carbon use efficiency in soil through methods comparison. *Soil Biology & Biochemistry*, 128, 79-88.
- Geyer, K. M., Kyker-Snowman, E., Grandy, A. S., & Frey, S. D. (2016). Microbial carbon use efficiency: accounting for population, community, and ecosystem-scale controls over the fate of metabolized organic matter. *Biogeochemistry*, 127(2-3), 173-188.
- Hagerty, S. B., Allison, S. D., & Schimel, J. P. (2018). Evaluating soil microbial carbon use efficiency explicitly as a function of cellular processes: implications for measurements and models. *Biogeochemistry*, 140(3), 269-283.
- Li, J., Wang, G., Mayes, M. A., Allison, S. D., Frey, S. D., Shi, Z., . . . Melillo, J. M. (2019). Reduced carbon use efficiency and increased microbial turnover with soil warming. *Global Change Biology*, 25, 900-910.

- Mahecha, M. D., Reichstein, M., Carvalhais, N., Lasslop, G., Lange, H., Seneviratne, S. I., . . . Cescatti, A. (2010). Global convergence in the temperature sensitivity of respiration at ecosystem level. *Science*, *329*(5993), 838-840.
- Schipper, L. A., Hobbs, J. K., Rutledge, S., & Arcus, V. L. (2014). Thermodynamic theory explains the temperature optima of soil microbial processes and high Q₁₀ values at low temperatures. *Global Change Biology*, *20*(11), 3578-3586.
- Schipper, L. A., Petrie, O. J., O'Neill, T. A., Mudge, P. L., Liáng, L. L., Robinson, J. M., & Arcus, V. L. (2019). Shifts in temperature response of soil respiration between adjacent irrigated and non-irrigated grazed pastures. *Agriculture, Ecosystems & Environment*, *285*, 106620.
- Sierra, C. A. (2012). Temperature sensitivity of organic matter decomposition in the Arrhenius equation: some theoretical considerations. *Biogeochemistry*, *108*(1-3), 1-15.
- Sinsabaugh, R. L., Moorhead, D. L., Xu, X., & Litvak, M. E. (2017). Plant, microbial and ecosystem carbon use efficiencies interact to stabilize microbial growth as a fraction of gross primary production. *New Phytologist*, *214*(4), 1518-1526.
- Steinweg, J. M., Plante, A. F., Conant, R. T., Paul, E. A., & Tanaka, D. L. (2008). Patterns of substrate utilization during long-term incubations at different temperatures. *Soil Biology & Biochemistry*, *40*(11), 2722-2728.
- Todd-Brown, K., Randerson, J., Post, W., Hoffman, F., Tarnocai, C., Schuur, E., & Allison, S. (2013). Causes of variation in soil carbon simulations from CMIP5 Earth system models and comparison with observations. *Biogeosciences*, *10*(3), 1717-1736.
- Tucker, C. L., Bell, J., Pendall, E., & Ogle, K. (2013). Does declining carbon-use efficiency explain thermal acclimation of soil respiration with warming? *Global Change Biology*, *19*(1), 252-263.
- Wang, G., & Post, W. M. (2012). A theoretical reassessment of microbial maintenance and implications for microbial ecology modeling. *FEMS Microbiology Ecology*, *81*, 610-617.
- Wang, G., Post, W. M., Mayes, M. A., Frerichs, J. T., & Jagadamma, S. (2012). Parameter estimation for models of ligninolytic and cellulolytic enzyme kinetics. *Soil Biology and Biochemistry*, *48*, 28-38.
- Xu, X., Schimel, J. P., Janssens, I. A., Song, X., Song, C., Yu, G., . . . Thornton, P. E. (2017). Global pattern and controls of soil microbial metabolic quotient. *Ecological Monographs*, *87*(3), 429-441.
- Ye, J.-S., Bradford, M. A., Maestre, F. T., Li, F.-M., & García-Palacios, P. (2020). Compensatory thermal adaptation of soil microbial respiration rates in global croplands. *Global Biogeochemical Cycles*, *n/a*(*n/a*), e2019GB006507.

Ye, J. S., Bradford, M. A., Dacal, M., Maestre, F. T., & Garca-Palacios, P. (2019). Increasing microbial carbon use efficiency with warming predicts soil heterotrophic respiration globally. *Global Change Biology*, 25(10), 3354-3364.

Zhou, X., Xu, X., Zhou, G., & Luo, Y. (2018). Temperature sensitivity of soil organic carbon decomposition increased with mean carbon residence time: Field incubation and data assimilation. *Global Change Biology*, 24(2), 810-822.

REVIEWERS' COMMENTS:

Reviewer #1 (Remarks to the Author):

I hope that replacement of the term "intrinsic temperature sensitivity" with "model-derived temperature sensitivity" will serve to avoid some confusion by readers. I am not convinced that the results reached in the present study are widely applicable, as I am aware of other recent studies showing no evidence of thermal adaptation to soil warming. However, this manuscript has undergone enough scrutiny that the results for this particular experiment warrant publication. I have a few suggestions for minor revisions:

Line 107: typo: replace "modeling" with "model"

Line 222: insert "model-derived" in front of "temperature sensitivity"

Line 259: "recapitulating" doesn't seem like the correct word. Perhaps "indicating" would be better

Lines 341-343: rewrite to: "As the MEND model uses different response functions to represent the effects of soil pH, temperature, and moisture on various transformation processes (Table S5), the MEND model attempts to derive a Q10 that specifically reflects the microbial and enzymatic responses to temperature change." I think you need a little more humility here, as I have suggested with this wording. I remain skeptical that it really achieves this goal, but I understand that is the intent.

Line 370: Add two new sentences: "This evidence for thermal adaptation in the present study contrasts with a recent meta-analysis of soil warming experiments, which found few significant differences in the temperature sensitivity of soil respiration between control and warmed plots across biomes and only limited evidence of acclimation of soil respiration to experimental warming (ref 10). This area of research clearly warrants additional study to understand differences in reported results among studies."

The authors cite reference #10 (Carey et al. 2016) in several places regarding the importance of this topic, but they never discuss the main conclusions of that large meta-analysis. I think it is appropriate for the main conclusion of this study, presented in lines 341 – 370, to discuss that the results of the present study are quite different from the conclusion of that meta-analysis and that the topic clearly needs more study to understand these differences.

Responses to Comments from Reviewers

We have addressed the reviewers' comments and suggestions point-by-point. The original reviewer's comments are italicized and our responses to the reviewer's comments follow. The numbers of lines in the text are referred to the revised version, where corrections are tracked.

A. Responses to Reviewer #1 (Remarks to the Author):

A1.

I hope that replacement of the term “intrinsic temperature sensitivity” with “model-derived temperature sensitivity” will serve to avoid some confusion by readers. I am not convinced that the results reached in the present study are widely applicable, as I am aware of other recent studies showing no evidence of thermal adaptation to soil warming. However, this manuscript has undergone enough scrutiny that the results for this particular experiment warrant publication. I have a few suggestions for minor revisions:

Responses: Thank you very much for the complimentary comments and constructive suggestions.

A2.

Line 107: typo: replace “modeling” with “model”.

Responses: Thank you for pointing it out. We have replaced “modeling” with “model” in the revised manuscript (Line 102).

A3.

Line 222: insert “model-derived” in front of “temperature sensitivity”.

Responses: We agree with this revision. “model-derived” is inserted in front of “temperature sensitivity” in the revised manuscript (Line 217).

A4.

Line 259: “recapitulating” doesn't seem like the correct word. Perhaps “indicating” would be better.

Responses: We agree that “indicating” is better than “recapitulating” here. The “recapitulating” is replaced by “indicating” (Line 255).

A5.

Lines 341-343: rewrite to: “As the MEND model uses different response functions to represent the effects of soil pH, temperature, and moisture on various transformation processes (Table S5), the MEND model attempts to derive a Q_{10} that specifically reflects the microbial and enzymatic responses to temperature change.” I think you need a little more humility here, as I have suggested with this wording. I remain skeptical that it really achieves this goal, but I understand that is the intent.

Responses: We appreciate the constructive comments and suggestions. We rewrite the sentence as “As the MEND model uses different response functions to represent the effects of soil pH, temperature, and moisture on various transformation processes (Table S5), the MEND model attempts to derive a Q_{10} that specifically reflects the microbial and enzymatic responses to temperature change” (Line 337-340).

A6.

Line 370: Add two new sentences: “This evidence for thermal adaptation in the present study contrasts with a recent meta-analysis of soil warming experiments, which found few significant differences in the temperature sensitivity of soil respiration between control and warmed plots across biomes and only limited evidence of acclimation of soil respiration to experimental warming (ref 10). This area of research clearly warrants additional study to understand differences in reported results among studies.”

The authors cite reference #10 (Carey et al. 2016) in several places regarding the importance of this topic, but they never discuss the main conclusions of that large meta-analysis. I think it is appropriate for the main conclusion of this study, presented in lines 341 – 370, to discuss that the results of the present study are quite different from the conclusion of that meta-analysis and that the topic clearly needs more study to understand these differences.

Responses: We fully agree with the reviewer. The two sentences are added in Line 366-371 as “This evidence for thermal adaptation in the present study contrasts with a recent meta-analysis of soil warming experiments, which found few significant differences in the temperature sensitivity of soil respiration between control and warmed plots across biomes and only limited evidence of acclimation of soil respiration to experimental warming (ref 10). This area of research clearly warrants additional study to understand differences in reported results among studies”.